



# Analysis of temporal and spatial variability of atmospheric CO₂ concentration within Paris from the GreenLITE™ laser imaging experiment

Jinghui Lian[1], François-Marie Bréon[1], Grégoire Broquet[1], T. Scott Zaccheo[2], Jeremy Dobler[3,a], Michel Ramonet[1], Johannes Staufer[4,b], Diego Santaren[1], Irène Xueref-Remy[5,b], and Philippe Ciais[1]

[1]Laboratoire des Sciences du Climat et de l'Environnement (LSCE), IPSL, CEA-CNRS-UVSQ, Université Paris-Saclay, Gif-sur-Yvette, France
[2]Atmospheric and Environmental Research, Lexington, Massachusetts, United States
[3]Spectral Sensor Solutions LLC, Fort Wayne, Indiana, United States
[4]Thales, Labège, France
[5]Institut Méditerranéen de Biodiversité et d'Ecologie marine et continentale (IMBE), Aix Marseille Université, CNRS, IRD, Avignon Université, Aix-en-Provence, France
[a]formerly at: Harris Corporation, Fort Wayne, Indiana, United States
[b]formerly at: Laboratoire des Sciences du Climat et de l'Environnement (LSCE), IPSL, CEA-CNRS-UVSQ, Université Paris-Saclay, Gif-sur-Yvette, France

Correspondence to: Jinghui Lian (Jinghui.Lian@lsce.ipsl.fr)

**Abstract**. In 2015, the Greenhouse gas Laser Imaging Tomography Experiment (GreenLITE™) measurement system was deployed for a long-duration experiment in the center of Paris, France. The system measures near-surface atmospheric CO₂ concentrations integrated along 30 horizontal chords ranging in length from 2.3 km to 5.2 km and covering an area of 25 km² over the complex urban environment. In this study, we use this observing system together with six conventional in-situ point measurements and the WRF-Chem model coupled with two urban canopy schemes (UCM, BEP) at a horizontal resolution of 1 km to analyze the temporal and spatial variations of CO₂ concentrations within the Paris city and its vicinity for the 1-year period spanning December 2015 to November 2016. Such an analysis aims at supporting the development of CO₂ atmospheric inversion systems at the city scale. Results show that both urban canopy schemes in the WRF-Chem model are capable of reproducing the seasonal cycle and most of the synoptic variations in the atmospheric CO₂ point measurements over the suburban areas, as well as the general corresponding spatial differences in CO₂ concentration that span the urban area. However, within the city, there are larger discrepancies between the observations and the model results with very distinct features during winter and summer. During winter, the GreenLITE™ measurements clearly demonstrate that one urban canopy scheme (BEP) provides a much better description of temporal variations and horizontal differences in CO₂ concentrations than the other (UCM) does. During summer, much larger CO₂ horizontal differences are indicated by the GreenLITE™ system than both the in-situ measurements and the model results, with systematic east-west variations.

## 1 Introduction

Urban areas account for almost two-thirds of global energy consumption and more than 70% of carbon emissions (IEA, 2008). Human activities, such as fossil fuel burning (Duren and Miller, 2012) and cement production (Wang et al., 2012) produce a net increase of atmospheric CO₂ concentration within and downwind the location of emissions. Over the years, many deployable instruments have been used to measure the urban atmospheric CO₂ concentrations, including (i) ground-based monitoring networks in e.g., Paris (Xueref-Remy et al., 2018), Indianapolis (Davis et al., 2017), Los Angeles (Feng et al., 2016), Washington, DC (Mueller et al., 2017), Boston (Sargent et al., 2018); (ii) airborne campaigns conducted in e.g., Colorado (Graven et al., 2009),


London (Font et al., 2015); (iii) existing space-based measurements, e.g., GOSAT (Hamazaki et al., 2004), OCO-2 (Crisp et al., 2008, 2015) and (iv) future satellites with imaging capabilities, e.g., OCO-3 (Elderling et al., 2019), GeoCarb (Moore et al., 2018) and CO2M (Buchwitz, 2018). These observations are used or could be used for estimating emissions of $CO_2$ over large cities using atmospheric inverse modeling, or to detect emission trends if atmospheric data are collected over sufficiently long periods. High-

accuracy continuous in-situ ground-based measurements of $CO_2$ concentrations, using the Cavity Ring-Down Spectroscopy (CRDS) technology, have been used in previous urban atmospheric inversion studies for the quantification of $CO_2$ emissions of large cities (Bréon et al., 2015; Staufer et al., 2016; Lauvaux et al., 2016; Feng et al., 2016; Boon et al., 2016; Sargent et al., 2018). However, many in-situ stations are needed to accurately capture the $CO_2$ emission budget of a large city (Wu et al., 2016), which requires high cost and efforts to set-up and maintain. The sparseness of $CO_2$ concentration sampling sites limits the ability of inversions to

estimate the large spatial and temporal variations of the $CO_2$ emissions within the city, even though high-resolution emission inventories are available (e.g. AIRPARIF, 2013).

New concepts and technologies are desirable for a full sampling of atmospheric $CO_2$ concentrations within a city. These concepts may rely on moderate precision but low-cost sensors that could be deployed at many sites for a high spatial density sampling (Wu et al., 2016; Arzoumanian et al., 2019). An alternative to in-situ point measurements is a remote sensing system based on the

spectroscopic techniques which could provide long-path measurements of atmospheric trace gases over extended areas of interest. An example of this type of approach is the differential optical absorption spectroscopy (DOAS). It has been applied to monitor atmospheric air pollutions such as nitrogen dioxide ($NO_2$) and aerosol in a complex urban environment (Edner et al., 1993). A novel laser absorption spectroscopy based system for monitoring greenhouse gases was developed by Harris Corporation, in partnership with Atmospheric and Environmental Research (AER). This system, known as the GreenLITE™, consists of

continuously operating laser sources and reflectors separated by a few kilometers. Both data collection and data processing components are based on the Intensity Modulated Continuous Wave (IM-CW) measurement technique, which is described in detail in Dobler et al. (2017). This instrument provides estimates of the average $CO_2$ concentrations along the lines of site defined by the paths between the laser based transceivers and a set of retroreflectors. The path between a transceiver and a reflector is referred to as a "chord". The GreenLITE™ system was developed and deployed as part of several field campaigns over the past several years

(Dobler et al., 2013; Dobler et al., 2017). These field tests have included extended operations at industrial facilities, and have shown that the system is capable of identifying and spatially locating point sources of greenhouse gases ($CO_2$ and $CH_4$) within a test area with relatively homogeneous background (~1 km$^2$). In the context of the 21st Conference of Parties to the United Nations Framework Convention on Climate Change (COP 21), the GreenLITE™ system was deployed for a long-duration field test to measure the $CO_2$ concentrations along 30 horizontal chords ranging in length from 2.3 km to 5.2 km and covering an area of 25

km$^2$ over central Paris, France. The aim of this field campaign was to demonstrate the ability of GreenLITE™ to monitor the temporal and spatial variations of near-surface atmospheric $CO_2$ concentrations over the complex urban environment. In addition, these measurements may be used for post-deployment analysis of the $CO_2$ distribution with the ultimate goal of revealing the $CO_2$ emission distribution. As a first step, the objectives of this work are to assess the information content of the GreenLITE™ data, and to analyze the atmospheric $CO_2$ distribution and to characterize precisely the processes that lead to dilution and mixing of the

anthropogenic emissions, which can provide new insights compared to the present in-situ point measurement approaches due to a much wider coverage and spatial representativeness.

The collection of the GreenLITE™ atmospheric $CO_2$ measurements in Paris makes it possible to evaluate and potentially improve meteorological and atmospheric transport models coupled to $CO_2$ emission inventories. On the other hand, the modeling system is expected to provide interpretations of the temporal and spatial variations of the GreenLITE™ data, with the aim of supporting the

development of $CO_2$ atmospheric inversion systems at the city scale. Here we compare GreenLITE™ $CO_2$ data with simulations





performed with the Weather Research and Forecasting Model coupled with a chemistry transport model (WRF-Chem). The WRF-Chem model allows various choices of physics parameterizations and data assimilation methods for constraining the meteorological fields (Deng et al., 2017; Lian et al., 2018). Previous studies have shown that it is necessary to account for specific urban effects when modeling the transport and dispersion of $CO_2$ over complex urban areas such as Salt Lake City, UT and Los

Angeles, CA (Nehrkorn et al., 2013; Feng et al., 2016). Nevertheless, even when accounting for the specificity of the urban environment, uncertainties in the modeling of atmospheric transport is a challenge, and significant mismatches remain between modeled and measured concentrations that could be explained by transport biases, particularly at night, but also in terms of vertical mixing during the day.

In this study, we present the results from a set of 1-year simulations (from December 2015 to November 2016) of $CO_2$

concentrations over the Paris megacity based on the WRF-Chem model coupled with two urban canopy schemes at a horizontal resolution of 1 km. The simulated $CO_2$ concentrations are compared with observations from the GreenLITE™ laser system as well as with in-situ $CO_2$ measurements taken continuously at six stations located within the Paris city and its vicinity. The results are discussed in the context of the measurement capability of the GreenLITE™ laser system and the performances of the high-resolution WRF-Chem model for the transport of $CO_2$ over the Paris urban canopy. We focus on the impact of heterogeneous

patterns in city emissions at 1 km resolution and urban atmospheric meteorology on the temporal and spatial variations of $CO_2$ concentrations.

This paper is organized as follows: Section 2 provides more details about the GreenLITE™ deployment in conjunction with the in-situ $CO_2$ monitoring network in Paris. The WRF-Chem modeling framework and model configurations are presented in Section 3. In Section 4, we evaluate the performance of the WRF-Chem simulations based on the analyses of the temporal and spatial

patterns of observed and modeled $CO_2$ concentrations. Discussions and conclusions are given in Section 5.

## 2 The observation network

### 2.1 In-situ measurements

Since 2010, a growing network of three to six in-situ continuous $CO_2$ monitoring stations has been established in the Ile-de-France (IdF) region in coordination with research projects (e.g., Bréon et al., 2015; Xueref-Remy et al., 2018). These observations are

used to understand the variability of atmospheric $CO_2$ concentrations, and the aim is to improve the existing bottom-up $CO_2$ emission inventories using them as a top-down constraint through atmospheric inverse modeling. The stations are equipped with high-precision $CO_2/CO/CH_4$ analyzers and installed on the rooftops or on towers to increase the area of representativity. All instruments have been regularly calibrated against the WMO cylinders (WMO-$CO_2$-X2007 scale) (Tans et al., 2011).

The locations of the stations are given in Table 1a and are shown in Figure 1. Four stations are located within the peri-urban area:

SAC and OVS sites are located about 22 km and 26 km southwest of Paris center with the respective sampling heights of 15/100 m and 20 m above the ground level (AGL). The other two sites are located at the north (AND) and north-east (COU) edges of the Paris urban area in a mixed urban-rural environment with inlets at 60 m and 30 m AGL respectively. These four peri-urban stations are complemented by in-situ continuous measurements at two urban stations: one in the Cité des Sciences et de l'Industrie (CDS) and one in the former Pierre and Marie Curie University (now Sorbonne University, also called Jussieu; JUS). The inlets for each

of the sensors are placed approximately 34 m and 30 m AGL respectively. The JUS station is on the roof of a building close to ventilation outlets and may be influenced by this and other localized sources of $CO_2$. The JUS site was only measuring $CO_2$ continuously from January to April 2016, and from September 16th 2016 to the end of the time period of this study. The location of this observation network was chosen a priori to allow the analysis of the gradient due to the Paris emissions when the winds



blow from the south-west and north-east directions, which corresponds to the prevailing winds in the region (Bréon et al., 2015; Staufer et al. 2016; Xueref-Remy et al, 2018).

**2.2 The GreenLITE™ campaign over Paris**

The GreenLITE™ system was deployed in Paris in November 2015 as a proof-of-concept demonstration for the COP 21 conference.

This system used two transceivers coupled with 15 retroreflectors to measure the $CO_2$ concentrations along 30 intertwined lines (chords) of 2.3-5.2 km length covering an area of 25 km$^2$ over the center of Paris. Each transceiver used two fiber-coupled distributed feedback lasers to generate an absorption line at a wavelength of 1571.112 nm and an offline with significantly lower absorptions (nominally 1571.061 nm). The experimental design and layout examined in this study are given in Table 1b and are illustrated in Figure 1. The two transceivers were located on the roof of the lower of the two Montparnasse buildings (T1) and on

the Jussieu tower (T2) located near the JUS in-situ instrument. They were chosen to be high enough, at 50.3 and 86.8 m AGL respectively in order to have a clear line of sight to the retroreflectors which were installed on rooftops around the city with heights varying from 16.8-50.4 m AGL. For this implementation, each transceiver scanned to the retroreflectors in sequence and made a transmission measurement of each chord with a period of four minutes. The experiment lasted from November 2015 to November 2016 although there were some down times of either the transceivers or some reflectors.

Preliminary analysis shows that the original GreenLITE™ $CO_2$ concentrations have a slow seasonal drift of approximately +/-5 ppm in comparisons to both the nearby in-situ measurements (Figure S1) and simulations with the CHIMERE-ECMWF transport configuration presented in Staufer et al. (2016). Therefore, a calibration method was developed by AER (Zaccheo et al., 2019) for addressing observed slowly drifting biases between the GreenLITE™ prototype system and the two in-situ sensors (CDS and JUS) that are near the GreenLITE™ chords. Unlike in-situ point measurement systems, there is no known method for directly traceable

calibration of long open-path systems to the WMO mole fraction scale used as an international standard for atmospheric $CO_2$ monitoring (Tans et al., 2011). The approach taken uses an adjustment to the offline wavelength to align the GreenLITE™ raw data from all chords with the absolute median values of two in-situ hourly data sets (CDS and JUS) over 4-day windows during the measurement period. Analyses have shown that this adjustment of an offset during a 4-day moving average has no significant impact on the higher frequency and chord-to-chord variations.

In order to enable the data to be compared to hourly in-situ observations and WRF-Chem outputs, hourly means are computed from the 4-minute GreenLITE™ data after applying the calibration approach described above but with two criteria: i) the number of samples per hour is greater than 3, and ii) the standard deviation (std) of the samples within the relevant hour is smaller than 10 ppm. Data that do not meet the above criteria, being only about 1.06 % of the total, are considered invalid and are excluded from further analysis.

**3 Modeling framework**

**3.1 WRF-Chem model setup**

A set of high-resolution simulations of atmospheric $CO_2$ concentrations is performed with the WRF-Chem V3.9.1 online coupled with the diagnostic biosphere Vegetation Photosynthesis and Respiration Model (VPRM) (Mahadevan et al., 2008; Ahmadov et al., 2007, 2009). The simulations are carried out over the period spanning September 2015 to November 2016, in which the first

three months are considered as a spin-up period. Three one-way nested domains are employed with the horizontal grid resolution of 25, 5 and 1 km, covering Europe (Domain 01), Northern France (Domain 02) and the IdF region (Domain 03) respectively (Figure S2). The meteorological initial and lateral boundary conditions are imposed using the ERA-Interim global re-analyses with





0.75 °×0.75 °horizontal resolution and 6 hourly intervals (Berrisford et al., 2011). We nudge the 3D fields of temperature and wind to the ERA-Interim reanalysis in layers above the planetary boundary layer (PBL) of the outer two domains using the grid nudging option of WRF. We also assimilate observation surface weather station data (ds461.0) and upper-air meteorological fields (ds351.0) from the Research Data Archive at the National Center for Atmospheric Research (https://rda.ucar.edu/datasets/ds351.0/;

https://rda.ucar.edu/datasets/ds461.0/) using a nudging technique (the surface analysis nudging and observation nudging options of WRF, which are described in details in Lian et al., 2018). Details regarding the model configurations used in this study are summarized in Table 2.

The urban canopy parameterization is a critical element in reproducing the lower boundary conditions and thermal structures, which are of vital importance for accurate modeling of the transport and dispersion of $CO_2$ within the urban areas. We therefore

pay special attention, in this study, to examine the impact of two available urban canopy models on WRF transport results, namely the single-layer Urban Canopy Model (UCM) (Chen et al., 2011) and the multilayer urban canopy model Building Effect Parameterization (BEP) (Martilli et al., 2002). This study does not assess the multilayer urban parameterization BEP+BEM (BEP combined with the Building Energy Model (BEM)) (Salamanca et al., 2010) since this parameterization focuses on the impact of heat emitted by air conditioners, which are not commonly used in Paris. This study uses 34 vertical layers in WRF-UCM with the

model top pressure at 100 hPa, and 15 layers arranged below 1.5 km with the first layer top at approximately 19 m AGL. To take full advantage of the WRF-BEP model, it is necessary to have a fine vertical discretization close to the surface so that this configuration is discretized with 44 vertical layers wherein 25 of them distributed within 1.5 km with the lowest level top being at around 3.8 m AGL. Previous sensitivity tests indicate that different physical schemes in the WRF-Chem model lead to mean differences of 2-3 ppm on the simulated $CO_2$ concentrations over Paris, whereas the various urban canopy schemes lead to much

larger differences. Thus in this study, we do not conduct tests of sensitivity to other physical schemes, and both experiments use the following parameterizations in all the modeling domains: WSM6 microphysics scheme (Hong and Lim, 2006), RRTM longwave radiation scheme (Mlawer et al., 1997), Dudhia shortwave radiation scheme (Dudhia, 1989), MYJ PBL scheme (Janjić, 1990, 1994), Eta Similarity surface layer scheme (Janjić, 1996), Unified Noah land-surface scheme (Chen and Dudhia, 2001). The Grell 3D ensemble cumulus convection scheme (Grell and Dévényi, 2002) is applied for Domain 01 only in both experiments.

**3.2 CO₂ simulations**

### 3.2.1 Anthropogenic CO₂ fluxes

Anthropogenic $CO_2$ fluxes within the IdF region are imposed using the AirParif inventory for the year 2010 at spatiotemporal resolutions of 1 km and 1 h (AIRPARIF, 2013). This inventory is based on various anthropogenic activity data, emission factors and spatial distribution proxies, which are described in details in Bréon et al. (2015). It provides maps and diurnal variations for

five typical months (January, April, July, August, and October) and three typical days (a weekday, Saturday and Sunday) to account for the seasonal, weekly and diurnal cycles of the emissions (see Figure 3, Bréon et al., 2015). Hourly $CO_2$ emissions from fossil fuel $CO_2$ sources outside the IdF region are taken from the inventory of the European greenhouse gas emissions with a spatial resolution of 5 km (updated in October 2005) developed by the Institute of Economics and the Rational Use of Energy (IER), University of Stuttgart, under the CarboEurope-IP project (http://www.carboeurope.org/).

Both inventories are adapted to the WRF-Chem model for the period of simulation (2015.09-2016.11). Moreover, we scale these two data sets to account for annual changes in emission between the base years and simulation timeframe. This is accomplished by rescaling the maps with the ratio of the annual budgets of national $CO_2$ emissions for the countries within the domain between the base year 2005 for IER and 2010 for AirParif and the year of simulation (2015/2016), taken from the Global Carbon Atlas (GCA) (http://www.globalcarbonatlas.org/en/CO2-emissions). Finally, we interpolate the emissions to the WRF-Chem grids



following the principle of mass conservation. Note that for the point sources such as stacks, industries and mines, $CO_2$ emissions are put in a single grid cell corresponding to their locations. Figure 2 shows the spatial distribution of the total $CO_2$ emissions for a weekday in March over the IdF region at the resolution of $1 \times 1$ km$^2$. It can be seen that there is a large spatial variability of $CO_2$ emissions ranging from 0 to more than 600 gCO$_2$/m$^2$/day in this area and the largest emissions are concentrated over the Greater Paris area, taking up about 50% of the emitted $CO_2$.

Based on the analysis of sectoral specific fossil fuel $CO_2$ emissions over the IdF region by Wu et al. (2016), we group the detailed sectoral AirParif emissions into five main sectors, namely building (43%), energy (14%), surface traffic (29%), aviation-related surface emissions (4%), and all other sectors (10%), where the percentages in parenthesis express the relative contribution of each sector to the total. All emissions are injected in the first model layer. Distinct $CO_2$ tracers are used for each of the five main sectors in the transport model to record their distinct $CO_2$ atmospheric signature. Figure 3 shows averages at the monthly scale of emissions below the GreenLITE™ chords for those different sectors. It illustrates that $CO_2$ emissions have a large seasonal cycle, mostly due to the residential heating (the "building" sector) which is strongly driven by variations of the atmospheric temperature. Figure 3 also reveals lower emissions for those chords (TX and R01-03) in the west of Paris than those in the other quadrants.

### 3.2.2 Biogenic $CO_2$ fluxes

Biogenic $CO_2$ fluxes are simulated with the VPRM model forced by meteorological fields simulated by WRF, and coupled to the atmospheric transport. VPRM uses the simulated downward shortwave radiation and surface temperatures, along with the vegetation indices (EVI, LSWI) derived from the 8-day MODIS Surface Reflectance Product (MOD09A1) and four parameters for each vegetation category (PAR0, λ, α, β) that are optimized against eddy covariance flux measurements over Europe collected during the Integrated EU project "CarboEurope-IP" (http://www.bgc-jena.mpg.de/bgc-processes/ceip/). The land cover data used by the VPRM (see Figure S3) are derived from the 1-km global Synergetic Land Cover Product (SYNMAP, Jung et al., 2006) reclassified into 8 different vegetation classes (Ahmadov et al., 2007, 2009).

Figure 4a shows the spatial distribution of daytime-averaged (06-18 UTC) $CO_2$ biogenic flux (NEE with a negative sign indicating net $CO_2$ uptake by the vegetation surface) in June 2016. The model simulates negative values of NEE (uptake of more than 5 gCO$_2$/m$^2$/day) over most of the region with the exception in urban areas where the values are assigned to zero. Figure 4b shows the mean diurnal cycles of NEE for 12 calendar months and for 8 vegetation classes used in the VPRM over Domain 03. The magnitude of NEE is highly dependent on the vegetation types, although the diurnal cycles are similar across these vegetation types. From November to January, the VPRM estimates within the IdF region show a small diurnal cycle and a positive NEE explained by ecosystem respiration exceeding gross primary productivity. One exception to positive wintertime NEE is for evergreen trees which, according to the VPRM model, sustain enough gross primary productivity to keep a negative daytime NEE throughout the year. The model shows large $CO_2$ uptake between late spring and early summer. Note that the seasonal cycle of crops, which dominates over the IdF region, is somewhat different from that of forests, with a NEE that decreases after the harvest in June/July, this crop phenology signal is being driven by the MOD09A1 data. Grasses also have a shorter uptake period than the other vegetation types, with a positive NEE as early as August.

### 3.2.3 Initial and lateral boundary conditions for $CO_2$

Initial and lateral boundary conditions for $CO_2$ concentration fields used in the WRF-Chem model are taken from the 3-hourly fields of the CAMS global $CO_2$ atmospheric inversion product (Chevallier, 2017a, 2017b) with a horizontal resolution of $3.75\,°\times1.90\,°$ (longitude $\times$ latitude) and 39 vertical levels between the surface and the tropopause.



## 4 Results

### 4.1 Time series and general statistics

The continuous $CO_2$ concentration measurement network in the IdF region provides an invaluable opportunity for model validation and data interpretation. In this work, the correlation coefficient, root-mean-square error (RMSE) and mean bias error (MBE)

metrics are first used to compare the performance of the WRF-Chem model with respect to the observed $CO_2$ concentrations from both the GreenLITE™ laser system and in-situ continuous stations. In order to compare them with the GreenLITE™ measurements, the modeling results are sampled and integrated along the chord lines, accounting for their positions and heights. For the in-situ point measurements, we simply use the $CO_2$ values from the 1-km WRF grid cell that contains the observation location.

Table 3, together with Figure S4 in the supplement, shows the statistics of all the hourly differences between the observed and

modeled $CO_2$ concentrations and the hourly afternoon differences (11-16 UTC), from December 2015 to November 2016 using the two model configurations (UCM, BEP). The results presented in the Taylor diagrams (Figure S4) are based on the full year of data and the seasonal statistics are summarized in Table 3. In general, the model performance is better during the afternoon, both in terms of correlation and RMSE, than it is for the full day. These results are consistent with previous findings that show the model has little skills at reproducing the $CO_2$ fields during the nighttime due to poor representation of vertical mixing during

nighttime conditions, and in the morning due to inadequate depiction of PBL growth (e.g. Bréon et al., 2015; Boon et al. 2016). Given the better performance of the WRF-Chem model in the afternoon, we focus the following analyses on $CO_2$ concentrations acquired during this period of the day only.

The other significant feature is that the UCM model shows a large positive bias (8.7-19.6 ppm) with respect to the observations within the city during autumn and winter. In contrast, the statistics for the BEP model results compared to the observations are

significantly better with clear improvements in the correlation and substantial decreases both in RMSE and MBE. It is well known that the low atmosphere is, on average, more stable in winter than in summer (Gates, 1961). As a consequence, a significant fraction of the emitted $CO_2$ remains close to the surface, so that its atmospheric concentrations is, in winter, highly sensitive to local fluxes and variations in vertical mixing, especially in the complex urban areas. The statistics are highly dependent on the choice of the urban canopy model, which strongly suggests that the large UCM model-measurement mismatches in winter are linked to

difficulties in modeling the vertical mixing within the urban canopy. It is worth noting that $CO_2$ concentrations are better reproduced by both UCM and BEP in the spring, with correlations that fluctuate between 0.51 and 0.82 across stations. Both models show lower correlations during summer (0.45-0.63). These lower values are mostly due to the smaller variability of the concentration rather than a higher measurement-model mismatch. Moreover, the UCM and BEP also have comparable performances at peri-urban areas while the BEP is slightly better at some suburban sites as shown by the statistics. The smallest

errors (both in terms of RMSE and bias) are found at Saclay with a measurement inlet that is well above the sources at 100 m AGL (SAC100).

The statistics shown in Table 3 and Figure S4 also indicate the ability of the models to reproduce the $CO_2$ at two urban in-situ stations (JUS & CDS) and the averages of the GreenLITE™ measurements over the T1 and T2 chord ensembles, calculated separately. In general, the model performance is similar for the two types of urban measurements, whereas the performance for

urban measurements is slightly inferior to that of the suburban (both in terms of RMSE and correlation). The correlations with observations are better for T1 and T2 than for the two urban in-situ sites, which may be due to the fact that T1 and T2 represent an average over a wide area, and is then less sensitive to local unresolved sources than the in-situ measurements. The RMSE with the BEP model is within the range of 4.5 to 9.6 ppm for T1 which is in some respect superior to those of JUS and CDS. In terms of the MBE, the values of T1 are similar with those of CDS, while the BEP simulation reveals an underestimation of $CO_2$ for T2 and

JUS, with a negative bias of up to 5.2 ppm.





Figure 5 shows time series of modeled $CO_2$ against daily afternoon mean GreenLITE™ observations (11-16 UTC). Again, it clearly illustrates that the UCM model overestimates the $CO_2$ concentrations close to the surface within the city during winter. The BEP model effectively reproduces the seasonal cycle, as well as most synoptic variations of the atmospheric $CO_2$ measurements. Note that the UCM model-observation discrepancies for T2 are much smaller than those of T1 as the transceiver T2 is 36.5 m higher in altitude, whereas such a difference in modeled $CO_2$ between T1 and T2 is not obvious for the BEP model.

### 4.2 Analyze co-variations of $CO_2$ spatial difference with wind

In this section, we analyze the spatial variations of the $CO_2$ concentrations measured at the in-situ stations, provided by the GreenLITE™ system and simulated by the WRF-Chem model. The analysis of spatial differences rather than individual values should strongly reduce the signature of the large-scale pattern due e.g. boundary conditions, and better highlight that of the Paris emissions (Bréon et al. 2015), which makes it possible to further evaluate some characteristics of the model and the measurement data.

### 4.2.1 In-situ measurement

We analyze the horizontal differences between pairs of in-situ stations as a function of wind speed and direction, expecting a larger concentration at the downwind station with respect to the upwind station, in this region of high emission. For wind fields, we use the ECMWF high-resolution operational forecasts (HRES) linearly interpolated at the hourly resolution, and extracted at a height of around 25 m AGL (https://www.ecmwf.int/en/forecasts/datasets/set-i) as a proxy for all stations located within the IdF region even though the wind fields might be slightly different between the Paris city and its surroundings due to the impact of urban structure (e.g. the difference in wind speed is less than 0.5m/s, as shown in Lian et al., 2018). Furthermore, the hourly afternoon $CO_2$ data are classified into the wind classes with a bin-width of 1 m/s for wind speed and 11.25 ° for wind direction. Figure 6 shows the patterns of the observed and modeled $CO_2$ concentration differences between pairs of in-situ stations, averaged accounting for the wind classes. The std values of the $CO_2$ concentration differences for each wind class are shown in Figure S5. Figure 6a shows the observed and modeled $CO_2$ horizontal differences between AND and COU, two suburban stations located to the north of the Paris city. One expects that stations downwind of sources of emissions would have a higher $CO_2$ concentration than those upwind so that the sign of the difference should vary with the wind direction. For this pair of sites (AND and COU), both the model and observations show the expected pattern with a similar amplitude. The values of RMSE and MBE are 4.53 and -0.14 ppm respectively for the BEP model, implying a slightly better performance than the UCM model (6.34 and -0.47 ppm respectively).

Figure 6b and 6c show similar figures but for the $CO_2$ differences of (COU-SAC) and (CDS-SAC). The Paris city is located between both pairs of stations when the wind is roughly from the north-east or from the south-west directions. Both COU and SAC are located outside of the city and show a pattern with fairly symmetric positive and negative values. Conversely, CDS is in the Paris city, within an urban environment, and is strongly affected by significant urban emissions from its surroundings. As a consequence, the CDS-SAC differences in concentration are mostly positive for all wind sectors, with the exception of very specific wind conditions (low winds in the 45 °north-east sector). The wind speed also has a strong influence on the differences. The $CO_2$ difference signal and its variability (std) are generally larger for smaller wind speeds. The model plots (second and third rows) illustrate that the models reproduce well the expected cross-city upwind-downwind differences in $CO_2$ concentrations. In term of signal amplitude, the BEP model is also in better agreement with the observations than the UCM model, which is particularly true for the std values shown in Figure S5.





Conversely, both models fail to reproduce the wind-related pattern of the observed CDS-JUS difference (Figure 6d). These observed differences do not show any upwind-downwind patterns and are mostly negative, which can be expected since JUS is close to the city center where strong emissions impact the concentration, whereas CDS is in the middle of a park and is therefore less affected by emissions from its surroundings. The modeled pattern is dominated by the simple upwind-downwind structure and

it is very much different from the observed values, especially when the wind is out of west to south-west, where the model values are positive and the observed differences are strongly negative. This model-measurement discrepancy is likely the result of a poor description of the emissions in the city center that are not well reproduced by the 1-km resolution inventory with periodic temporal profiles. It may also indicate that the complex urban structure and morphology, such as buildings and street canyons affect the energy budget and atmospheric transport, all of which lead to fine-scale (sub-kilometer) $CO_2$ concentration features that cannot be

captured by the WRF-Chem model at a 1-km horizontal resolution. The in-situ point measurement may then not be representative of the average within the larger area (1 km$^2$) that is simulated by the model.

The analysis of the in-situ point measurement differences within and around Paris, together with the simulations, indicates that the model reproduces both the general structure and the amplitude of the cross-city differences in $CO_2$ concentrations and the $CO_2$ difference in the Paris surroundings, but that it mostly fails to simulate $CO_2$ differences between the two stations located in the

inner city.

### 4.2.2 GreenLITE™ measurement

One expects that the GreenLITE™ principle, that provides averaged $CO_2$ concentrations along the chord lines, is less affected by the local unresolved sources of $CO_2$ emissions than the in-situ point measurements. Meanwhile, the wide spatial coverage of the GreenLITE™ system is expected to provide additional information about $CO_2$ spatial variations within the Paris city. In this section,

we focus on the spatial variation of $CO_2$ concentration measured with the GreenLITE™ system. As a first step, we analyze the distribution of the absolute values of the observed hourly afternoon $CO_2$ difference between all pairs of chords for each month together with their simulated counterparts shown in Figure 7.

We first focus on the winter period (December to February). During that period, the median value of the measured T1 inter-chord range is mostly on the order of 2 ppm. That of T2 is somewhat larger, on the order of 3-4 ppm with some excursions up to 9 ppm.

The two models UCM and BEP show very large differences. Whereas BEP simulates spatial variations that are of the right order of magnitude compared to the GreenLITE™ data, those of UCM are much larger. Thus, the GreenLITE™ measurements provide clear information that favors the BEP model versus the UCM. During the winter period, there is little vertical mixing which leads to large vertical gradients in $CO_2$ concentrations close to the surface. The two models differ in their representations of this mixing which leads to large differences in the modeled $CO_2$ concentrations. Figure S6 shows that the UCM model reproduces a much

larger vertical gradient in $CO_2$ concentrations close to the surface, a few tens meters above the emissions than the BEP model does during afternoon (11-16 UTC). The differences are not as large higher up, neither are they further downwind of the emissions as the vertical gradient is then smoother as a result of mixing.

During the summer period, solar insulation generates more instability and the convection generates vertical mixing that limits the horizontal gradients. Both models indicate an inter-chord range of less than a few ppm. Conversely, the GreenLITE™ data indicate

much larger values, of 3-4 ppm (the median) for T1 and even larger for T2. Further analysis indicates that this spatial variation is mostly systematic, i.e. that some chords are consistently lower or higher than the in-situ values. At this point, there are three hypotheses:

• H1 The spatial differences of T1 and T2 are true features linked to fine-scale spatial variations of the emissions between the west and east part of Paris, that are underrepresented or not included in the emission inventory;





- H2 The models fail in the description of $CO_2$ concentrations within the Paris city, as the analysis of JUS and CDS in-situ measurements has shown;

- H3 There is a chord-dependent bias in some of the GreenLITE™ chords during the summer period.

To resolve this question, we look at the spatial difference between the in-situ sites within the city (JUS-CDS) during summer. Unfortunately, the JUS instrument was not working during the summer of 2016. Therefore, we use the JUS and CDS data over the summers from December 2015 to December 2018 (Figure 7c). In general, the modeled $CO_2$ concentration differences between pairs of in-situ stations are larger than the modeled inter-chord range of the GreenLITE™ system. During the summer, the observed absolute differences between JUS and CDS are only of a few ppm (the median is on the order of 2 ppm during July and August). These observations indicate that the spatial differences of $CO_2$ between these two sites within the Paris city are much smaller during the summer than during the winter, and tend to support the modeling results, which would undermine the assumptions H1 and H2.

However, these two stations do not sample the western part of Paris that is less densely populated with a higher fraction of green areas. The in-situ observations do not fully rule out, therefore, the possibility of an impact of the emission spatial structure.

In order to get further insights into the characteristics of $CO_2$ spatial variations within the Paris city, we analyze the $CO_2$ differences with the consideration of the anthropogenic $CO_2$ emissions shown in Figure 2 and Figure 3. We therefore group the 15 chords from T1 into three parts according to both their geographic locations and the amounts of anthropogenic $CO_2$ emissions averaged along the chords: the western, middle and eastern parts consist of reflectors R01, R02, R03, reflectors R06, R07, R08, and reflectors R13, R14, R15 respectively overlying three different regions within Paris. Figure 8 shows the co-variations of the GreenLITE™ observed and modeled $CO_2$ spatial difference with winds. The std values of the $CO_2$ concentration differences for each wind class are shown in Figure S7.

In Figure 8b and 8c, we show the east-west and the middle-west differences, where the $CO_2$ anthropogenic emissions in the western part are systematically lower than the other two regions, the observed $CO_2$ concentrations in the middle and east are on average higher than the west. The patterns of observed $CO_2$ difference are characterized by positive values no matter where the wind blows. The $CO_2$ differences reproduced by the model are positive in the southwest direction, however, it shows a nearly opposite pattern with those from observations when the wind is from the northeast. A plausible explanation for this is that the influence of km-scale anthropogenic emissions over different parts of Paris on the observed $CO_2$ concentration has a greater effect than the atmospheric transport and dispersion of the fluxes over the period of study.

Figure 8a shows similar figures but for the east-middle difference. There is a better measurement-model agreement than for Figure 8b and 8c. Indeed, the spatial variations of $CO_2$ concentrations show, as expected, negative values over upwind directions and positive values over downwind directions both for the observation and the model. According to the inventory, the two Paris areas that are covered by the set of chords used here have similar anthropogenic emissions. As a consequence, the overall $CO_2$ concentration difference, as shown in Figure 8a, is then better linked to the impact of atmospheric transport.

We therefore conclude that the pattern of $CO_2$ concentration difference is consistent with winds only over the areas with similar anthropogenic emissions. In other terms, if we compare $CO_2$ concentrations of the chords overlaying different level of emissions, the model may be insufficient in accurately modulating the dispersion of $CO_2$ emissions, the ventilation and dilution effects at such a high urban microscale resolution.





## 5 Summary and Conclusions

In this study, we use conventional in-situ together with novel GreenLITE™ laser measurements for an analysis of the temporal and spatial variations of the $CO_2$ concentrations within the Paris city and its vicinity. The analysis also uses 1 km-resolution WRF-Chem model coupled with two urban canopy schemes, for the 1-year period from December 2015 to November 2016.

The results have shown very distinct features during winter and summer:

During the winter, the emissions within the city are the highest, mainly due to households heating, and the vertical mixing is low. This combination leads to large temporal, vertical and horizontal variations of $CO_2$ concentrations. The GreenLITE™ measurements are used to clearly demonstrate that the BEP model provides a much better description of the $CO_2$ fields within the city than the UCM model does. On the other hand, both models show similar performances in the city surrounding.

During the summer, the emissions are lower (by a factor of roughly two compared to the cold season) and the sun-induced convection makes the vertical mixing much faster than in winter. For this period, both the in-situ measurements and the modeling indicate that, during the afternoon, the spatial differences are limited to a few ppm. Much larger spatial differences are indicated by the GreenLITE™ system, with systematic east-west variations. This is not yet fully understood.

This study stresses the difficulty in reproducing the atmospheric $CO_2$ concentration within the city because of our inability to

represent the detailed spatial structure of the emission and because of the sensitivity of the concentration to the strength of vertical mixing. There are strong indications that the uncertainty on the vertical mixing is much larger than the uncertainty on the emissions so that atmospheric concentration measurements within the city can hardly be used to constrain the emission inventories.

### Author contribution

JL, FMB, GB and PC contributed to the design and implementation of the research. JL, FMB, GB, PC, TSZ and JD contributed

to the analysis and interpretation of the results. TSZ, JD, MR and IXR performed the measurements. JS and DS contributed to model input preparation. JL and FMB took the lead in writing the manuscript with input from all authors.

### Code/Data availability

All data sets and model results corresponding to this study are available upon request from the corresponding author.

### Competing interests

The authors have no competing interests to declare.

### Acknowledgements

This work is supported by the Ph.D. program funded by the IDEX Paris-Saclay, ANR-11-IDEX-0003-02 together with Harris Corporation. We would like to thank Harris Corporation and the management of Atmospheric and Environmental Research, Inc. for their support of these ongoing analyses. We acknowledge the support of Francois Ravetta (LATMOS/IPSL) and Sorbonne

University, for the installation of the GreenLITE™ system on the Jussieu Campus. Thanks also to Paris Habitat, Elogie and Montparnasse ICADE for providing the locations to install the transceivers and reflectors. Finally, thanks to Marc Jamous at Cité des Sciences et de l'Industrie (CDS), to Cristelle Cailteau-Fischbach (LATMOS/IPSL), to OVSQ, and to LSCE/RAMCES technical staff for the maintenance of the in-situ monitoring network, coordinated by Delphine Combaz.



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

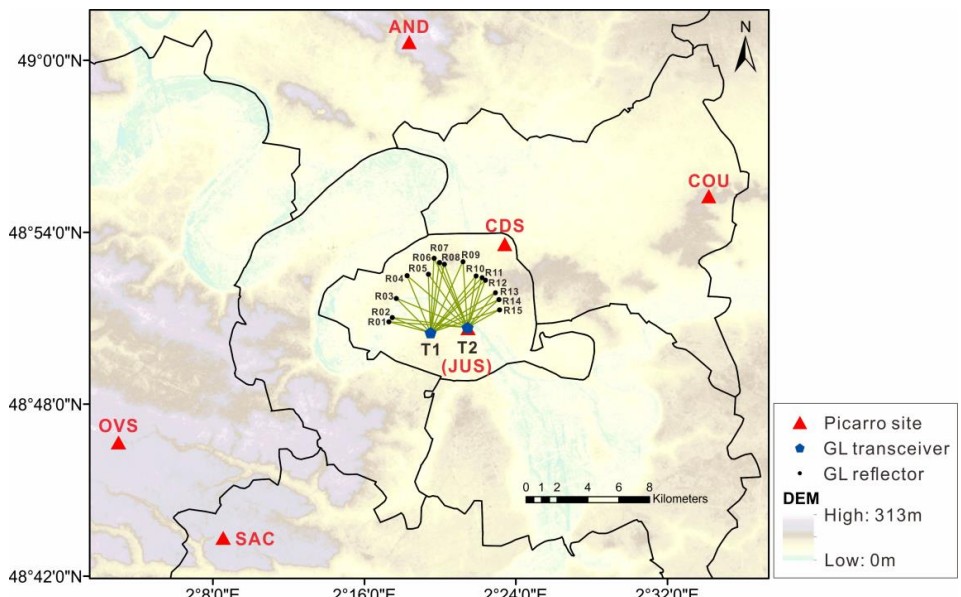

**Figure 1: Distribution of in-situ $CO_2$ measurements and the GreenLITE™ laser system.** The Paris city is located within the inner line, but the urban area extends over a larger surface, very roughly within the Greater Paris area (including Paris and the three administrative areas that are around Paris called "Petite Couronne" in French, see Figure S1). The Ile-de-France region covers an area that is larger than the domain shown here. (Data sources: the ASTER Global Digital Elevation Model (GDEM) Version 2 data are available at https://lpdaac.usgs.gov/products/astgtmv002/; the administrative division map of the Ile-de-France region is available at https://www.data.gouv.fr/en/datasets/geofla-departements-idf/, same for Figure 2, 4, S3)

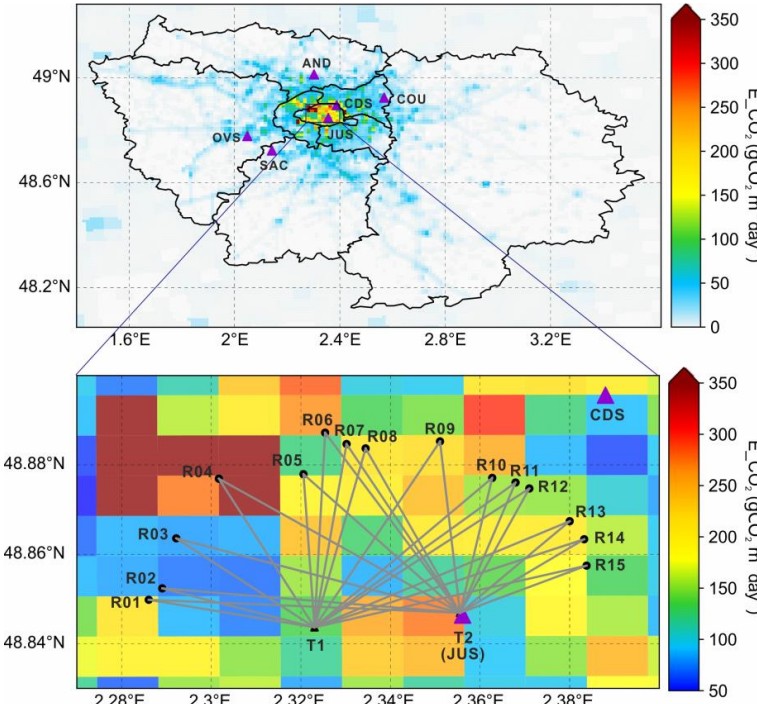

10  **Figure 2: Total $CO_2$ emissions for a weekday in March 2016.** The top figure shows the 1-km emissions over the IdF region together with the in-situ measurement stations. The bottom figure is a high-resolution zoom of the inner Paris area and shows the emissions together with the GreenLITE™ chords and two urban in-situ measurement stations.





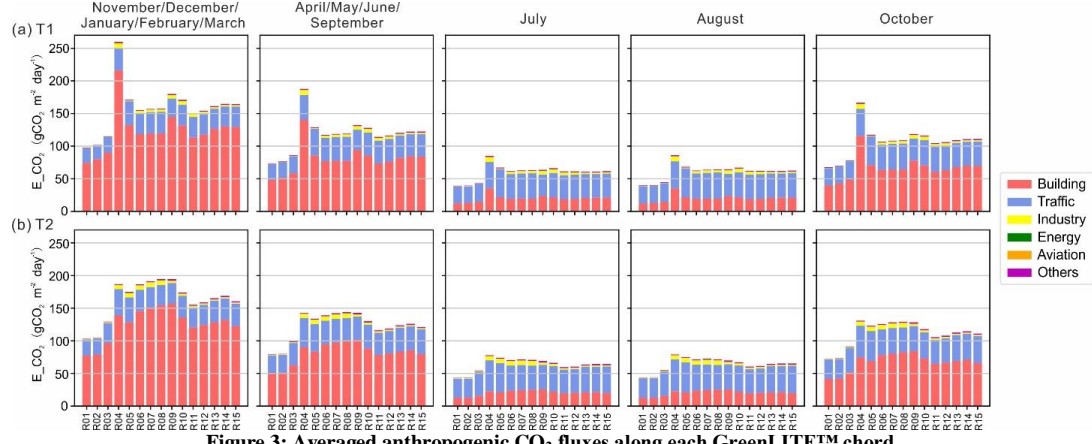

**Figure 3: Averaged anthropogenic CO₂ fluxes along each GreenLITE™ chord.**

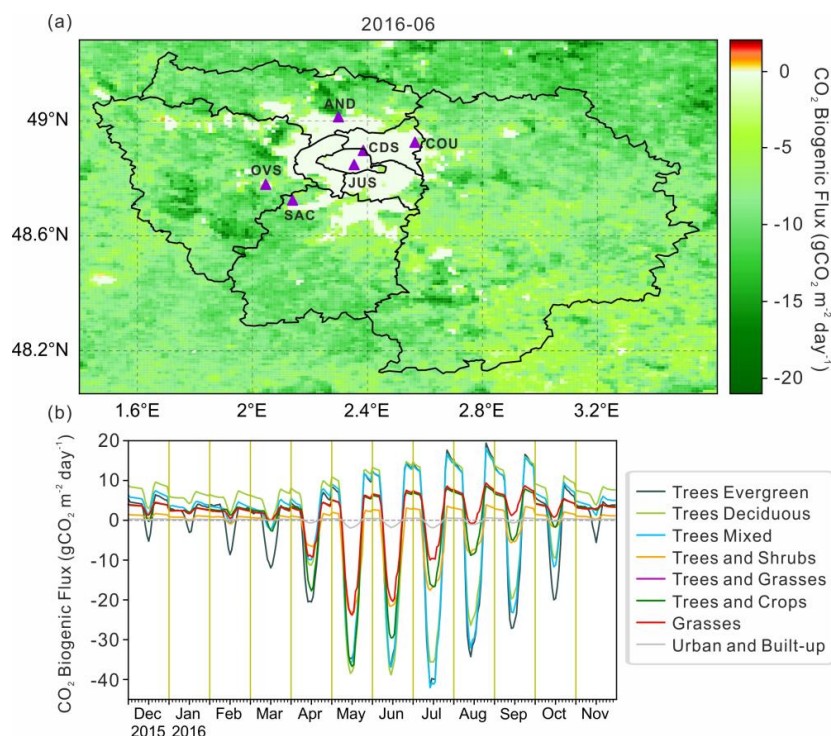

5    **Figure 4: (a) Daytime (06-18 UTC) average of CO₂ biogenic flux (NEE) in June 2016; (b) Mean diurnal cycles of CO₂ biogenic flux (NEE) for 12 calendar months and for 8 vegetation classes used in VPRM over Domain 03.**

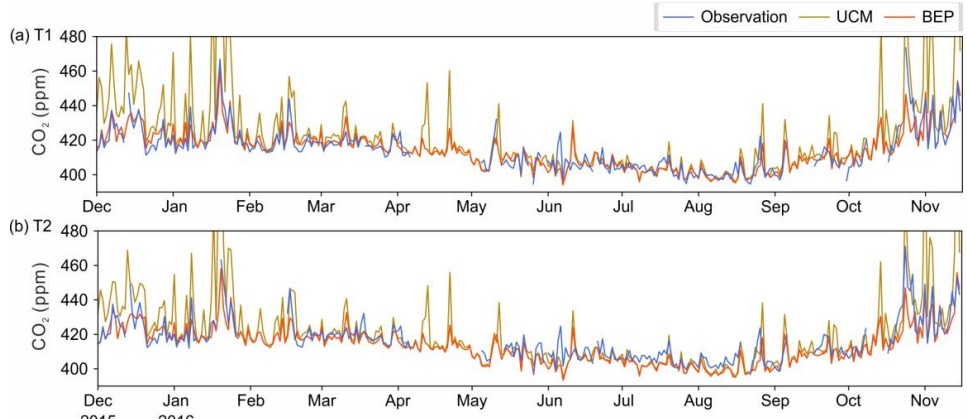

**Figure 5: Time series of the GreenLITE™ observed and modeled averaged CO₂ concentrations during afternoon (11-16 UTC) for the (a) T1 and (b) T2 chord ensembles.**

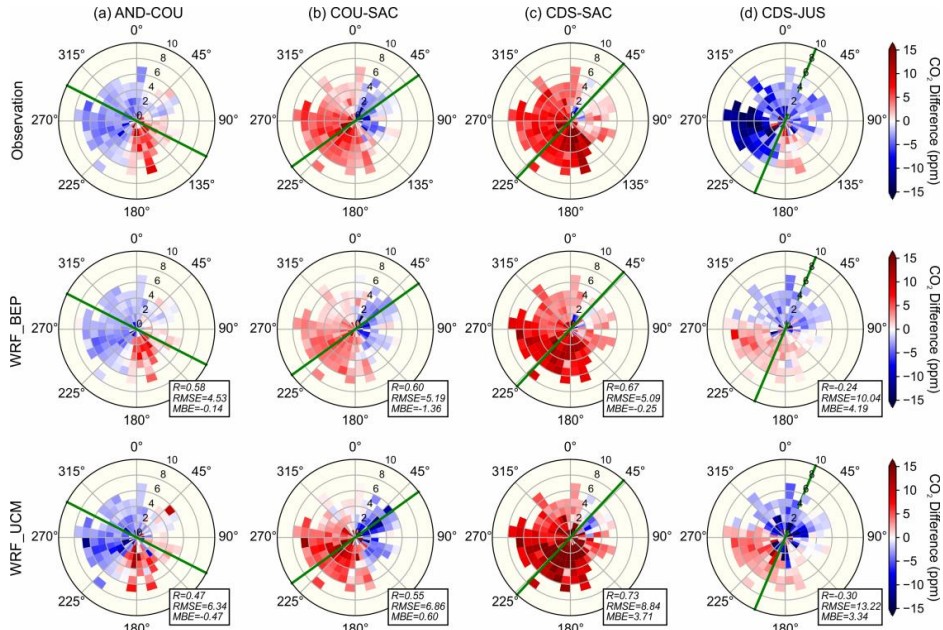

**Figure 6: Spatial differences in CO₂ concentration between two stations of the in-situ network, averaged over sets of situation corresponding to bins of wind speed and direction. Only the afternoon (11-16UTC) data are used. The top row shows the observations, whereas the other two rows show the two simulations (UCM, BEP). The green line indicates the direction defined by two in-situ stations. The statistics of hourly values of observed and modeled CO₂ concentration difference are shown in the box.**



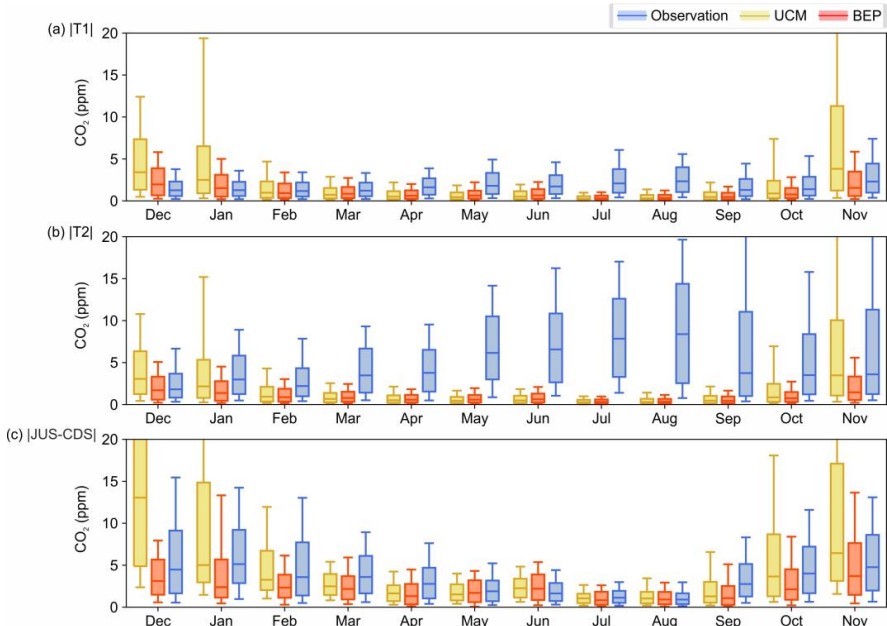

**Figure 7: Distribution of the GreenLITE™ observed and modeled absolute CO$_2$ concentration differences between all pairs of chords for (a) T1 and (b) T2 from December 2015 to November 2016. (c) Distribution of the observed and modeled absolute CO$_2$ concentration differences between JUS and CDS from December 2015 to December 2018. The midpoint, the box and the whiskers represent the 0.5 quantile, 0.25/0.75 quantiles, and 0.1/0.9 quantiles respectively. Note that only the afternoon data (11-16 UTC) are used in the analysis.**

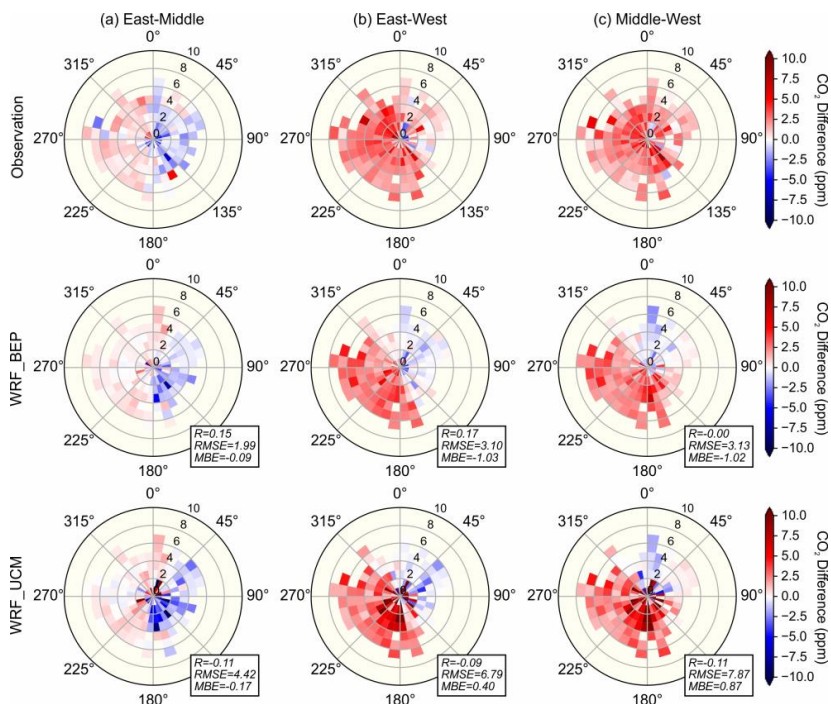

**Figure 8: Spatial differences in CO$_2$ concentration between (a) east-middle, (b) east-west and (c) middle-west parts of the GreenLITE™ T1 measurement, averaged accounting for wind speed and direction. Only the afternoon (11-16UTC) data are used. The top row shows the observations, whereas the other two rows show the two simulations (UCM, BEP). The statistics of hourly values of observed and modeled CO$_2$ concentration difference are shown in the box.**



**Table 1. Information about $CO_2$ observation stations used in this study.**

**(a) In-situ stations**

| Site | Acronym | Latitude (°) | Longitude (°) | Height AGL (m) |
|------|---------|--------------|---------------|----------------|
| Jussieu | JUS | 48.8464 | 2.3561 | 30 |
| Cité des Sciences | CDS | 48.8956 | 2.3880 | 34 |
| Andilly | AND | 49.0126 | 2.3018 | 60 |
| Coubron | COU | 48.9242 | 2.5680 | 30 |
| OVSQ | OVS | 48.7779 | 2.0486 | 20 |
| Saclay | SAC | 48.7227 | 2.1423 | 15/100 |

**(b) The GreenLITE™ system**

| | | R01 | R02 | R03 | R04 | R05 | R06 | R07 | R08 | R09 | R10 | R11 | R12 | R13 | R14 | R15 |
|---|---|-----|-----|-----|-----|-----|-----|-----|-----|-----|-----|-----|-----|-----|-----|-----|
| Chord | T1 | 2.80 | 2.67 | 3.17 | 4.02 | 3.81 | 4.84 | 4.59 | 4.53 | 5.06 | 4.72 | 4.88 | 4.93 | 4.94 | 4.93 | 4.71 |
| Length (km) | T2 | 5.11 | 4.91 | 5.00 | 5.17 | 4.30 | 5.00 | 4.59 | 4.38 | 4.28 | 3.40 | 3.37 | 3.30 | 2.90 | 2.74 | 2.39 |
| Height | R | 50.4 | 41.7 | 18.3 | 28.1 | 19.7 | 20.8 | 24.5 | 25.9 | 16.9 | 28.8 | 29.7 | 24.7 | 21.8 | 16.8 | 23.6 |
| AGL (m) | T | T1: 50.3; T2: 86.8 | | | | | | | | | | | | | | |





**Table 2. A summary of WRF-Chem configurations used in this study.**

| Option | | Setting |
|---|---|---|
| Simulation Periods | | 2015.09.01~2016.11.30 |
| Horizontal Resolution | | 25 km (Domain 01), 5 km (Domain 02), 1 km (Domain 03) |
| Boundary & Initial Conditions | Meteorology | ERA-Interim reanalysis data (0.75 °×0.75 °, 6 hourly) |
| | $CO_2$ concentration | LMDZ_CAMS (3.75 °×1.895 °, 3 hourly) |
| Nudging | | Grid nudging + Surface nudging + Observation nudging (NCEP operational global observation surface data (ds461.0) and upper-air data (ds351.0)) |
| Flux | Anthropogenic emissions | IER inventory for 2005 (5 km, outside IdF) + AirParif inventory for 2010 (1 km, within IdF) rescaled for 2015-2016 using national budgets from the GCA |
| | Biogenic NEE | VPRM (online coupling) |
| Physics Schemes | Microphysics | WSM6 scheme |
| | Cumulus convection | Grell 3D ensemble scheme only in Domain 01 |
| | Longwave radiation | RRTM scheme |
| | Shortwave radiation | Dudhia scheme |
| | PBL | MYJ scheme |
| | Surface layer | Eta Similarity scheme |
| | Vegetated land surface | Unified Noah land-surface model |
| | Urban land surface | UCM (34 vertical levels wherein 15 below 1.5 km) |
| | | BEP (44 vertical levels wherein 25 below 1.5 km) |





**Table 3.** Seasonal statistics for observed and modeled $CO_2$ concentrations for two urban canopy schemes (UCM, BEP) and periods of the day (all hourly data, hourly afternoon data) from December 2015 to November 2016. DJF denotes December-January-February, MAM denotes March-April-May, JJA denotes June-July-August and SON denotes September-October-November.

**(a) Correlation coefficient**

|  |  | T1 | | T2 | | JUS 30m | | CDS 34m | | SAC 15m | | SAC 100m | |
|---|---|---|---|---|---|---|---|---|---|---|---|---|---|
|  |  | UCM | BEP | UCM | BEP | UCM | BEP | UCM | BEP | UCM | BEP | UCM | BEP |
| All hourly | DJF | 0.68 | 0.67 | 0.58 | 0.67 | 0.62 | 0.63 | 0.49 | 0.51 | 0.68 | 0.77 | 0.59 | 0.73 |
|  | MAM | 0.52 | 0.56 | 0.52 | 0.61 | 0.39 | 0.39 | 0.48 | 0.53 | 0.64 | 0.71 | 0.64 | 0.67 |
|  | JJA | 0.63 | 0.61 | 0.60 | 0.60 | NA | NA | 0.52 | 0.55 | 0.68 | 0.72 | 0.59 | 0.63 |
|  | SON | 0.55 | 0.58 | 0.57 | 0.63 | 0.47 | 0.46 | 0.55 | 0.54 | 0.57 | 0.65 | 0.64 | 0.70 |
| Hourly afternoon (11-16 UTC) | DJF | 0.79 | 0.83 | 0.70 | 0.79 | 0.68 | 0.65 | 0.65 | 0.59 | 0.65 | 0.86 | 0.65 | 0.86 |
|  | MAM | 0.67 | 0.81 | 0.69 | 0.79 | 0.51 | 0.60 | 0.71 | 0.78 | 0.77 | 0.81 | 0.81 | 0.82 |
|  | JJA | 0.46 | 0.47 | 0.45 | 0.46 | NA | NA | 0.52 | 0.55 | 0.57 | 0.63 | 0.49 | 0.49 |
|  | SON | 0.73 | 0.83 | 0.71 | 0.82 | 0.55 | 0.73 | 0.65 | 0.75 | 0.77 | 0.83 | 0.74 | 0.82 |

**(b) Root-mean-square error (RMSE. Unit: ppm)**

|  |  | T1 | | T2 | | JUS 30m | | CDS 34m | | SAC 15m | | SAC 100m | |
|---|---|---|---|---|---|---|---|---|---|---|---|---|---|
|  |  | UCM | BEP | UCM | BEP | UCM | BEP | UCM | BEP | UCM | BEP | UCM | BEP |
| All hourly | DJF | 28.26 | 11.23 | 19.38 | 11.03 | 40.96 | 14.43 | 28.84 | 13.63 | 8.82 | 7.42 | 7.47 | 6.64 |
|  | MAM | 18.91 | 11.77 | 14.85 | 9.84 | 25.89 | 14.42 | 18.24 | 12.23 | 8.78 | 7.86 | 7.85 | 7.74 |
|  | JJA | 9.98 | 10.33 | 10.13 | 10.09 | NA | NA | 12.11 | 11.00 | 11.48 | 11.49 | 7.14 | 7.20 |
|  | SON | 32.94 | 20.06 | 25.23 | 18.11 | 43.50 | 24.22 | 29.57 | 20.27 | 13.82 | 13.20 | 9.46 | 8.97 |
| Hourly afternoon (11-16 UTC) | DJF | 31.82 | 5.98 | 23.79 | 6.68 | 42.31 | 10.08 | 33.75 | 9.61 | 8.14 | 5.33 | 7.08 | 4.92 |
|  | MAM | 7.84 | 4.47 | 6.69 | 5.12 | 9.17 | 6.11 | 7.27 | 4.79 | 5.75 | 4.55 | 5.11 | 4.47 |
|  | JJA | 7.07 | 5.99 | 7.51 | 7.25 | NA | NA | 7.26 | 5.46 | 5.86 | 4.06 | 5.04 | 4.56 |
|  | SON | 31.87 | 9.57 | 28.39 | 10.45 | 42.50 | 13.09 | 32.29 | 12.01 | 9.72 | 6.50 | 8.20 | 6.46 |

**(c) Mean bias error (MBE. Unit: ppm)**

|  |  | T1 | | T2 | | JUS 30m | | CDS 34m | | SAC 15m | | SAC 100m | |
|---|---|---|---|---|---|---|---|---|---|---|---|---|---|
|  |  | UCM | BEP | UCM | BEP | UCM | BEP | UCM | BEP | UCM | BEP | UCM | BEP |
| All hourly | DJF | 12.99 | -0.36 | 6.75 | -2.97 | 14.24 | -3.85 | 12.09 | -0.11 | 0.96 | -0.89 | -1.09 | -1.62 |
|  | MAM | 6.28 | 1.21 | 1.11 | -3.32 | 8.65 | -0.12 | 4.94 | -0.62 | 0.03 | -1.53 | -1.30 | -2.59 |
|  | JJA | 1.25 | 0.97 | -2.50 | -3.68 | NA | NA | 1.77 | 0.74 | -3.71 | -4.38 | -1.69 | -2.72 |
|  | SON | 14.06 | -0.83 | 5.33 | -6.20 | 17.70 | -1.39 | 8.99 | -3.05 | -0.64 | -3.98 | -0.27 | -2.01 |
| Hourly afternoon (11-16 UTC) | DJF | 17.37 | 0.99 | 12.99 | -0.90 | 13.55 | -5.24 | 19.61 | 2.69 | 3.51 | 1.74 | 0.59 | 0.21 |
|  | MAM | 2.59 | 0.59 | -0.72 | -2.71 | 0.58 | -2.36 | 2.91 | 0.52 | 3.22 | 1.59 | 2.08 | 0.46 |
|  | JJA | 0.66 | -0.89 | -2.65 | -4.09 | NA | NA | 1.85 | 0.06 | 3.14 | 1.62 | 1.13 | 0.17 |
|  | SON | 14.01 | -0.86 | 8.65 | -4.36 | 12.84 | -4.47 | 11.29 | -0.92 | 4.88 | 1.14 | 2.60 | 0.02 |