# Peer review of "Analysis of temporal and spatial variability of atmospheric CO2 concentration within Paris from the GreenLITETM laser imaging experiment"

_Atmospheric Chemistry and Physics, 2019_

## Referee Comment (RC1) · Anonymous Referee #1 · 5 Aug 2019

Review of Lian et al. (2019) Analysis of temporal and spatial variability of atmospheric CO2 concentration within Paris from the GreenLITETM laser imaging experiment

Lian et al. describe the application of a long open-path spectroscopy technique for the measurement of CO2 mixing ratios above a complex urban canopy, which could influence existing emissions estimates at the city scale. They compare the data measured using the GreenLITETM system with fixed-site CO2 measurements within the same urban environment, and contrast the results against two urban canopy schemes within the WRF-Chem model.

It is a well written paper which I would recommend for publication in ACP. The content

of the paper, which covers greenhouse gas measurements with a possible climate change impact, is relevant to the journal and its readers.

General comments:

The authors acknowledge that calibration of long open-path spectroscopy techniques is difficult. A separate paper (Zaccheo et al., 2019), detailing a new calibration procedure applied to the GreenLITE™ data, is referenced by the authors here. This calibration procedure appears to use the fixed-site installations within the city to calibrate the open-path data. Whilst the authors state this "has no significant impact on chord-to-chord variations", they do not discuss the potential implications of using point-source measurements to adjust area/path averaged measurements. Zaccheo et al. (2019) does go into more detail but considering this is a key element of the calibration procedure, I believe it needs some more attention here.

Font sizes in some figures could be larger. Some of the text is hard to read on a computer screen without zooming in.

The authors should address the following points in a revised manuscript:

Page 3, Line 30: What is meant by 15/100 m above ground level? I assume there are two sampling inlets? This is not made clear.

Page 4, Line 23: What is meant by "no significant impact" – significant in what way?

Page 4, Line 27: Why is a threshold of standard deviation < 10 ppm CO2 applied to the hourly data? Is this because CO2 is not expected to change by more than 10 ppm over the course of one hour? Is this justifiable?

Page 5, Line 19: Can you quantify "much larger differences"?

Page 5, Line 29: Typo - "details" should be "detail".

Page 6, Line 5: Consider "accounting for" rather than "taking up"?

Page 11, Line 9: Rephrase "city surrounding" to "areas surrounding the city", or similar.

Table 3: What are the colour scales showing? Better or worse values? This needs to be made more clear particularly because high correlation coefficient (red) is good but high RMSE (also red) is bad?

Figure 1: Some text is very small – a possible solution would be to refer the reader to the panel in Fig 2 in the caption and remove the chord labels. Also the caption refers to Figure S1 but this doesn't appear relevant to the text – the authors might mean Figure S2?

Figure 2: Does the caption need to state that these emissions are taken from an emissions inventory i.e. not measured or modelled.

Figure 3: See Figure 2.

Figure 4b: Is there some way of better highlighting that this is not a continuous time series of data? Perhaps either thicker/bolder lines or a gap between each monthly diurnal cycle.

Figure 5: The blue (observation) line is quite difficult to see on these plots.

Figure 7: y-axis titles should probably read "CO2 difference (ppm)" as in Figure 6.
* * *

---

## Referee Comment (RC2) · Anonymous Referee #2 · 6 Aug 2019

This paper describes urban measurements of CO2 by in-situ and by the novel open-path laser system "GreenLITE" with multiple reflectors and transceivers deployed in the Paris area. Observations are compared to high-resolution WRF-Chem simulations with a representation of CO2 fluxes from anthropogenic emissions and biosphere-atmosphere exchange. The paper is well written, and I recommend publishing after the following minor comments are addressed.

General comments:

For the WRF-Chem modeling CO2 emissions at annual and national scale for scaling the high spatial resolution emissions to the year of interest haven been taken from the

[Figure]

Global Carbon Atlas (GCA), however it is unclear what these data are based on (e.g. UNFCCC reporting, BP statistical reports, or other sources). The Global Carbon Atlas has some missing links in the "Data contributors" section making traceability of the emissions impossible. This needs to be clarified.

It is somewhat unclear how the statistics shown in Table 3 and Fig. S4 have been calculated for the GreenLITE vs. WRF-Chem measurements in section 4.1. Have the data from all chords related to e.g. T1 been combined and then the statistics is derived, or has each chord been treated independent and the resulting statistics shown in Table 3 and Fig. S4 reflect the average across all chords?

The discussion of the results in section 4.2.2 regarding the spatial gradients between different chords of the GreenLITE observations and the simulated counterparts, as well as the corresponding discrepancy between observations and model results should at least mention the potential impact of turbulent eddies and thermals. Those are likely to form in a convectively unstable atmosphere, i.e. during summer, and are unlikely to be represented properly in the MYJ PBL scheme (a local closure model) deployed in the WRF-Chem simulations (c.f. Xiao-Ming et al., 2010). Ref.: Hu, Xiao-Ming, John W Nielsen-Gammon, and Fuqing Zhang. 2010. "Evaluation of Three Planetary Boundary Layer Schemes in the WRF Model." Journal of Applied Meteorology and Climatology 49 (9): 1831–44. doi:10.1175/2010JAMC2432.1.

Specific comments:

P1 L36: I suggest replacing "have been used" with e.g. "have been or will be used" as you are refering also to future satellites.

P7 L21: please rephrase "low atmosphere", e.g. "lower part of the atmosphere"

P7 L38: "in some respect superior" this should be formulated clearer. What I see from Table 3 is that RMSE with the BEP model is always better for T1 than for JUS, and better for T1 than for CDS with one exception.

[Figure]

P8 L21, P8 L37, and P10 L19: please rephrase "The std values", e.g. "Standard deviations"

P9 L38: What is the difference between the first two of the three hypotheses? Is H2 meant to refer to only transport model deficiencies, excluding inaccuracies in emissions? This should be made clearer. Also it should be made clear at the end of section 4.2.2 which hypothesis remains the most probable one.
* * *

---

## Referee Comment (RC3) · Anonymous Referee #3 · 26 Aug 2019

Lian et al., present long open-path spectroscopy measurements for the City of Paris from December 2015 to November 2016 in conjunction with in-situ observations from towers in and around Paris as well as WRF-Chem simulated observations from two different urban canopy schemes. It is assumed that the authors are using the GreenLITE measurements along with the in-situ tower observations to discern which WRF-Chem urban canopy scheme can best represent vertical mixing and transport in urban areas.

My main concern with this article is that the specific objective/conclusions of the paper are unclear. The authors have conducted a lot of work analyzing data from many components, but it is uncertain as to whether they have drawn any solid conclusions.

[Figure]

The objective, which I assume is using GreenLITE and the in-situ observations to evaluate WRF-Chem urban canopy configurations, should be more clearly stated in the Introduction and the title of the paper should be changed.

Without a clear narrative, the paper mainly comes across as a presentation of data which is difficult to evaluate as a reader. I do not recommend that this work be accepted for publication in ACP without substantial revisions to clarify scientific objectives/conclusions.

I have the following suggestions regarding the technical components of the analysis.

Main Comments:

(1) The observational network: can you provide some indication as to areas in which the observations are most sensitive especially CDS, JUS, and any others that are situated in or adjacent to major sources/sinks? I understand that the authors use WRF-CHEM and not a Lagrangian approach so that footprints cannot be generated but having some understanding would help the comparison of GreenLITE and in-situ observations presented later in the paper since these two different types of measurements represent different spatial extents.

(2) GreenLITE campaign – can you provide more description than citing the Zaccheo paper as to how the GreenLITE observations are calibrated?

(3) WRF-Chem – Is this paper an analysis of WRF-Chem urban canopy models for cities like Paris (evaluated using GreenLITE and in-situ observations)? If so, please substantiate/provide reference for the claim that "previous sensitivity tests indicate that different physical schemes in the WRF-Chem model lead to mean differences of 2-3ppm on the simulated $CO_2$ concentrations over Paris, whereas the various urban canopy schemes lead to much larger differences." This seems like the motivation for much of the work presented within the paper, but I am not sure that this claim, if substantiated, holds true across most urban areas.

a. If this is not the focus, and the purpose is to use the meteorology to understand the variability of the measurements, then I believe the authors should pick a model and use it throughout the rest of the analysis. It seems (from Figure 5) that the BEP model is largely better. The rest of the analysis using UCM could be put in the supplementary information. As an aside, I do think that the authors could use the ensembles in a way that would help them draw some robust conclusions. Their ensembles provide some measure of the atmospheric transport and dispersion uncertainty which can be used to contextualize their comparison between GreenLITE and the in-situ observations (S1 and S4).

(4) Anthropogenic Fluxes – The use of the anthropogenic fluxes in the analysis should be reconsidered or better explained. For example, does IER have any temporal variability? If so, please explain. If not, the authors could consider scaling using published methods. The authors could use other emission products that have temporal variability if needed. The loss of spatial scale (e.g. going from 5-10km) seems less important than preserving some temporal structure in emissions. Other products are also also more recent and thus more represented of ex-urban fluxes which constitute a large portion of CO2 inflow to Paris. Could the authors also further explain "we interpolate the emissions to the WRF-Chem grids following the principle of mass conservation?" This is unclear in both its meaning and why it is important. As with the WRF-Chem comments, the authors could use an ensemble of anthropogenic emission products (those outside of Paris) to help contextualize the GreenLITE and in-situ observations in terms of emission uncertainty (refer to Martin et al., 2018).

(5) Biogenic Fluxes – The use of VPRM to represent the urban biosphere is an active area of research and there are lots of questions as to how well a biospheric model captures the urban biogenic emissions. When VPRM was optimized using flux data, were urban towers used to help parameterize the "urban" areas of Paris? The paper mentions that the western portion of Paris has much green space and thus biogenic sources might be important in this area of the city and impact the analysis. How was

Paris-VPRM (or VPRM) validated, e.g. comparison to in-situ data from towers outside of the city that are surrounded by vegetation (maybe OVS)? Has it been used in other studies? How does it vary as a function of time in comparison to the anthropogenic fluxes like what is shown in Figure 3?

(6) Results – (4.1) There are a lot of moving pieces in this analysis and it is hard to ascertain the main conclusions from the statistical analysis. Do you think that the uncertainties associated with the other components (e.g. anthropogenic emissions and vprm sources and sinks) would have changed some of these results especially during the growing seasons or per your analysis of the seasonality of the sectors? From the Table, it is unclear that BEP outperforms UCM for much of the year. (4.2.1) Why did you use the wind per ECMWF versus wind measurements at the upwind tower(s)? I am sure, on average, the ECMWF winds are similar to what is measured at the towers but since you are comparing hourly measurements, this may make a difference. Also, how much time does it take to traverse some of the towers that are farther apart (e.g. COU and SAC)? Did you compare observations from similar times or did you account for a lag in the measurements via travel time? As with 4.1, I am not sure what the main takeaway is from this analysis.

Minor Comments:

Be specific as to what model you are using. I think in most cases you are referring to WRF-Chem models but there are others too such as VPRM, etc.

Grammar should be checked in many places throughout the article to improve clarity. Examples include lines 34 through 36 (page 2), ∼10 (page 8).

Figures should be modified to improve clarity:

For example, Figure 1 should include a depiction of adjacent urban areas to show how remote AND, COU, OVS, and SAC are from ex-urban sources. This will help the reader know whether or not they sample "clean" air.

[Figure]

Figure 2 should include roads and other infrastructure in the second panel especially since the authors have made spent time discussing sectorial emissions. Also note (a) and (b) on Figure 2.

For Figure 4, zoom into similar area as in Figure 2 to show if VPRM is capturing urban biospheric flux which can significantly impact the urban fluxes especially their variability.

The authors should consider moving Figure 5 to the supplemental information. It doesn't provide much information, especially given the timeframe that makes it hard to see, expect to show that the UCM transport model yields extreme outliers in the winter.

For Table 3, explain by what criteria did you color code the Tables. It seems like the better models for the correlation coefficients have "red" shading where int the RMSE and MBE the colors are switched (aka blue is better while red is worse). I would remove "all hours" to make the table clearer - all hours not really needed.

---

## Author Comment (AC1) · 2 Oct 2019

We would like to thank Referee #1 for his/her thoughtful comments and detailed suggestions to our manuscript. In the following, we answer to the reviewer's comments and indicate the changes in the manuscript that were implemented as a consequence of the recommendations. The comments are in black and italic. Our answers are in blue and plain text.

**Anonymous Referee #1**

*Review of Lian et al. (2019) Analysis of temporal and spatial variability of atmospheric $CO_2$ concentration within Paris from the GreenLITE™ laser imaging experiment.*

*Lian et al. describe the application of a long open-path spectroscopy technique for the measurement of $CO_2$ mixing ratios above a complex urban canopy, which could influence existing emissions estimates at the city scale. They compare the data measured using the GreenLITE™ system with fixed-site $CO_2$ measurements within the same urban environment, and contrast the results against two urban canopy schemes within the WRF-Chem model.*

*It is a well written paper which I would recommend for publication in ACP. The content of the paper, which covers greenhouse gas measurements with a possible climate change impact, is relevant to the journal and its readers.*

We thank the reviewer for these very supportive comments.

*General comments:*

*The authors acknowledge that calibration of long open-path spectroscopy techniques is difficult. A separate paper (Zaccheo et al., 2019), detailing a new calibration procedure applied to the GreenLITE™ data, is referenced by the authors here. This calibration procedure appears to use the fixed-site installations within the city to calibrate the open-path data. Whilst the authors state this "has no significant impact on chord-to-chord variations", they do not discuss the potential implications of using point-source measurements to adjust area/path averaged measurements. Zaccheo et al. (2019) does go into more detail but considering this is a key element of the calibration procedure, I believe it needs some more attention here.*

We thank the reviewer for raising this question of data calibration. As he/she points out, the calibration procedure is described in some detail in (Zaccheo et al., 2019) so that we felt there is no need to go into the same level of detail. As it has been addressed in Zaccheo et al. (2019), while not desirable, it is often necessary to apply post-calibration corrections to such data to rectify residual differences between observation types. We acknowledge nevertheless that the reader may want to see more, and we therefore have provided more information about this calibration method as well as its limitation in the revised version of the manuscript:

"These slowly time-varying differences were most likely due to a slight systematic long-term drift in both the on- and off-line wavelengths as a function of continuous operations. Such drift may induce some non-linear impacts on the measured concentrations. It is therefore more appropriate to adjust the wavelengths rather than to apply a linear calibration to the retrieved concentrations. Unlike in-situ point measurement systems, there is no established method for calibration of long open-path systems to the WMO mole fraction scale used as an international standard for atmospheric $CO_2$ monitoring (Tans et al., 2011). Therefore, a bias correction method was developed by AER (Zaccheo et al., 2019) for addressing observed slowly drifting biases between the GreenLITE™ prototype system and the two in-situ sensors (CDS and JUS) that are near the GreenLITE™ chords. This method computed a time-varying adjustment to the offline

wavelength based on a non-linear optimization mechanism. This non-linear approach adjusts the GreenLITE™ offline wavelength considering not only the average values of hourly $CO_2$ concentrations at two in-situ stations, but also the corresponding average temperature, relative humidity, atmospheric pressure along the chord and an optimized online wavelength value during the measurement period. Finally, the median on- and off-line values over a 4-day window was used to recompute the GreenLITE™ data from all chords using a radiative transfer based iterative retrieval scheme based on the LBLRTM model (Clough et al., 2005). Even though this approach is not ideal as the two in-situ stations and the GreenLITE™ system do not sample the exact same area, it does provide a well-defined mechanism that reduces the systematic long-term biases with no significant impact on the chord-to-chord variations."

As most of our analyses focus on the spatial gradient on the concentrations, we feel that the important point of the calibration procedure is that it has no significant impact (and thus no significant uncertainty) on the spatial gradient between chords. (See below, our answers to the comment: Page 4, Line 23: What is meant by "no significant impact" – significant in what way?)

*Font sizes in some figures could be larger. Some of the text is hard to read on a computer screen without zooming in.*

This suggestion is well taken. We have increased the font size in all figures (see Figure 1 to 7 in the revised manuscript).

*The authors should address the following points in a revised manuscript:*

*Page 3, Line 30: What is meant by 15/100 m above ground level? I assume there are two sampling inlets? This is not made clear.*

Yes, the atmospheric $CO_2$ concentrations at SAC station are measured with two sampling inlets at 15 m and 100 m above ground level, on a tall tower at that location. This has been clarified in the text:

"OVS site is located about 26 km southwest of Paris center with the sampling height of 20 m above the ground level (AGL) on the top of a building. The SAC tall tower is located on the Plateau de Saclay (9.5 km southeast of OVS) with two air inlets placed at 15 m and 100 m AGL respectively."

In addition, we also changed "15/100" to "15 and 100" in Table 1.

*Page 4, Line 23: What is meant by "no significant impact" – significant in what way?*

We agree that the term "no significant impact" is an overly vague statement and should be clarified more precisely. We have added a figure in the revised supplement (Figure S2). It shows the distributions of the original and re-processed GreenLITE™ absolute $CO_2$ concentration differences between all pairs of chords for each transceiver. The differences between the medians of the inter-chord range of the re-processed and original data are within in the range of ±0.5 ppm for T1 and ±2 ppm for T2 with the respective yearly mean plus/minus one standard deviation of $0.04 \pm 0.16$ ppm for T1 and $0.48 \pm 0.43$ ppm for T2, which are relatively small. The modified text in the revised manuscript is as follows:

"Top panels in Figure S2 (a) and (b) show the distribution of the absolute values of the daily averaged $CO_2$ concentration difference between all pairs of chords for each transceiver before and after the calibration. The differences between the medians of the re-processed and original inter-chord range, shown in bottom panels, are within in the range of ±0.5 ppm for T1 and ±2 ppm for T2 with the respective yearly mean plus/minus one standard deviation of $0.04 \pm 0.16$ ppm for T1 and $0.48 \pm 0.43$ ppm for T2."

[Figure]

Figure S2: Distribution of the original and re-processed GreenLITE™ absolute $CO_2$ concentration differences between all pairs of chords for (a) T1 and (b) T2 from December 2015 to November 2016. The solid lines in top panels of (a) and (b) indicate the 0.5 quantile, and the shaded areas represent the 0.1 and 0.9 quantile intervals for original data in blue and re-processed data in red. The green line in bottom panels of (a) and (b) indicates the differences between the median values of the re-processed and original inter-chord range.

*Page 4, Line 27: Why is a threshold of standard deviation < 10 ppm $CO_2$ applied to the hourly data? Is this because $CO_2$ is not expected to change by more than 10 ppm over the course of one hour? Is this justifiable?*

The outlier detection for the 4-minute GreenLITE™ data is mainly based on the 3-sigma rule, which is used to remove the data outside three standard deviations from a mean in the positive direction. We have added a sentence in the main body of the revised manuscript and Figure S3 in the supplement to answer the reviewer's comment:

"The 10 ppm threshold was selected to be rough 3 times the typical standard deviation of the 4-minute measurements for any given chord within a one-hour period (Figure S3)."

For better clarity, we have also added the following statement in the supplement together with Figure S3.

"The outlier detection for the 4-minute GreenLITE™ data is mainly based on the 3-sigma rule, which is used to remove the data outside three standard deviations from a mean in the positive direction. Figure S3 (a) shows the frequency distribution of the standard deviations of the 4-minute $CO_2$ concentrations measured within one hour for one given chord (e.g. T2R08). Figure S3 (b) shows the three-sigma threshold (mean + $3\sigma$) of the standard deviations of the 4-minute measurements within a one-hour period for each chord. In general, the threshold varies between 6.5 ppm and 11.9 ppm from chord to chord. We therefore choose to use a uniform threshold value of 10 ppm to remove the outliers for all chords."

[Figure]

Figure S3: (a) Frequency distribution of the standard deviations of the 4-minute $CO_2$ concentrations measured within one hour for one chord (e.g. T2R08); (b) Three-sigma threshold (mean + $3\sigma$) of the standard deviations of the 4-minute measurements within a one-hour period for each chord.

*Page 5, Line 19: Can you quantify "much larger differences"?*

For better clarity, we have added the following sentence in the revised manuscript:

"In order to select an adequate model physical configuration for Paris, we carried out some preliminary sensitivity experiments to test the impact of different physical schemes on the simulated $CO_2$ concentrations. These tests use up to five different PBL schemes and two urban canopy schemes. The simulations were carried out for two months, including one winter month (January 2016) and one summer month (July 2016). These preliminary sensitivity results indicate that different PBL schemes in the WRF-Chem model lead to monthly average differences of 2-3 ppm on the simulated $CO_2$ concentrations over Paris, whereas the two different urban canopy schemes lead to much larger differences of 8-10 ppm. Thus in this study, we carried out the 1-year simulation with two different urban canopy schemes as they are sufficient to address the paper main question regarding the ability of a configuration of the WRF-Chem model to simulate the $CO_2$ atmospheric transport in an urban environment, but also to provide an estimate of the modeling uncertainty. All of the other physics options remained the same for the two experiments (Table 2)."

*Page 5, Line 29: Typo - "details" should be "detail".*

Correction made.

*Page 6, Line 5: Consider "accounting for" rather than "taking up"?*

Text changed as suggested.

*Page 11, Line 9: Rephrase "city surrounding" to "areas surrounding the city", or similar.*

Text changed as suggested. The modified text is as follows:

"On the other hand, both models show similar performances in the areas surrounding the city."

*Table 3: What are the colour scales showing? Better or worse values? This needs to be made more clear particularly because high correlation coefficient (red) is good but high RMSE (also red) is bad?*

We agree with the reviewer that the color scales in Table 3 can be misleading. The color only represents the values from minimum (blue) to maximum (red) in the cells instead of indicating the goodness of fit between model and observation. We have added the following text in the caption of Table 3 in order to clarify this issue:

"The color highlights the value in the cell with the minimum in blue, the median in white and the maximum in red. All other cells are colored proportionally."

*Figure 1: Some text is very small – a possible solution would be to refer the reader to the panel in Fig 2 in the caption and remove the chord labels. Also the caption refers to Figure S1 but this doesn't appear relevant to the text – the authors might mean Figure S2?*

For better clarity, we have added a second panel in Figure 1 and noted the previous Figure 1 as Figure 1a. Now, Figure 1a shows the distribution of in-situ $CO_2$ stations and the GreenLITE™ laser system without the chord labels. Figure 1b is a high-resolution zoom of the inner Paris area and shows the GreenLITE™ laser system layout in detail.

Corrected, thanks. It should refer to Figure S5 in the revised manuscript.

*Figure 2: Does the caption need to state that these emissions are taken from an emissions inventory i.e. not measured or modelled.*

Yes. We have added the following sentence into the caption to stress this point:

"Figure 2: Total $CO_2$ emissions, according to the AirParif inventory (within IdF) and the IER inventory (outside IdF), for a weekday in March 2016."

*Figure 3: See Figure 2.*

We have modified the caption:

"Figure 3: Averaged anthropogenic $CO_2$ fluxes along each GreenLITE™ chord according to the AirParif inventory."

*Figure 4b: Is there some way of better highlighting that this is not a continuous time series of data? Perhaps either thicker/bolder lines or a gap between each monthly diurnal cycle.*

As suggested we have used thicker lines between each monthly diurnal cycle in the revised Figure 4b.

*Figure 5: The blue (observation) line is quite difficult to see on these plots.*

We have changed the line colors to a sharper contrast between the observation and the model results. (PS: following the recommendation from Referee #3, we have moved this figure to the supplement as Figure S7 in the revised manuscript)

*Figure 7: y-axis titles should probably read "$CO_2$ difference (ppm)" as in Figure 6.*

Changed as suggested. This figure is now Figure 6 in the revised manuscript.

---

## Author Comment (AC2) · 2 Oct 2019

We would like to thank Referee #2 for his/her thoughtful comments and detailed suggestions to our manuscript. In the following, we answer to the reviewer's comments and indicate the changes in the manuscript that were implemented as a consequence of the recommendations. The comments are in black and italic. Our answers are in blue and plain text.

**Anonymous Referee #2**

*This paper describes urban measurements of $CO_2$ by in-situ and by the novel open-path laser system "GreenLITE" with multiple reflectors and transceivers deployed in the Paris area. Observations are compared to high-resolution WRF-Chem simulations with a representation of $CO_2$ fluxes from anthropogenic emissions and biosphere atmosphere exchange. The paper is well written, and I recommend publishing after the following minor comments are addressed.*

We thank the reviewer for his/her work and suggestions.

*General comments:*

*For the WRF-Chem modeling $CO_2$ emissions at annual and national scale for scaling the high spatial resolution emissions to the year of interest have been taken from the Global Carbon Atlas (GCA), however it is unclear what these data are based on (e.g. UNFCCC reporting, BP statistical reports, or other sources). The Global Carbon Atlas has some missing links in the "Data contributors" section making traceability of the emissions impossible. This needs to be clarified.*

We recognize that the sentence was insufficiently detailed. We have now modified the following statements with a supplement table (Table S1) to address the data sources and the corresponding references.

 "This is accomplished by rescaling the maps with the ratio of the annual budgets of national $CO_2$ emissions for the countries within the domain between the base year 2005 for IER and 2010 for AirParif and the year of simulation (2015/2016), taken from Le Quéré et al. (2018) (https://www.icos-cp.eu/GCP/2018. See also Table S1 in the supplement for details about original data sources)."

Table S1. National $CO_2$ emissions from fossil-fuel combustion and cement production for the countries within the WRF-Chem domain used in this study (unit: $MtCO_2$/yr). The data in the following table are taken from Le Quéré et al. (2018), available at https://www.icos-cp.eu/GCP/2018, last access: August 2019. (The use of data is conditional on citing the original data sources: data in black are from the CDIAC inventory (Boden et al., 2017), data in red are from the UNFCCC national inventory reports (UNFCCC, 2018), data in purple are from the BP Statistical Review of World Energy (BP, 2018). Cement emissions are updated from Andrews (2018))

|      | Austria | Belgium | France (including Monaco) | Germany | Italy (including San Marino) | Luxembourg | Netherlands | Spain | Switzerland | United Kingdom |
|------|---------|---------|---------------------------|---------|------------------------------|------------|-------------|-------|-------------|----------------|
| 2005 | 79.37   | 125.64  | 432.64                    | 867.22  | 495.23                       | 12.05      | 177.53      | 368.96| 45.78       | 570.00         |
| 2010 | 72.38   | 113.58  | 397.90                    | 833.68  | 424.87                       | 11.15      | 182.18      | 283.88| 45.05       | 512.21         |
| 2015 | 66.70   | 100.23  | 348.16                    | 797.08  | 355.48                       | 9.26       | 165.03      | 271.73| 38.74       | 422.66         |
| 2016 | 67.40   | 100.24  | 350.10                    | 801.75  | 350.32                       | 9.00       | 165.52      | 260.99| 39.20       | 398.55         |

References:

Le Quéré, C., Andrew, R. M., Friedlingstein, P., Sitch, S., Hauck, J., Pongratz, J., Pickers, P. A., Korsbakken, J. I., Peters, G. P., Canadell, J. G., Arneth, A., Arora, V. K., Barbero, L., Bastos, A., Bopp, L., Chevallier, F., Chini, L. P., Ciais, P., Doney, S. C., Gkritzalis, T., Goll, D. S., Harris, I., Haverd, V., Hoffman, F. M., Hoppema, M., Houghton, R. A., Hurtt, G., Ilyina, T., Jain, A. K., Johannessen, T., Jones, C. D., Kato, E., Keeling, R. F., Goldewijk, K. K., Landschützer, P., Lefèvre, N., Lienert, S., Liu, Z., Lombardozzi, D., Metzl, N., Munro, D. R., Nabel, J. E. M. S., Nakaoka, S., Neill, C., Olsen, A., Ono, T., Patra, P., Peregon, A., Peters, W., Peylin, P., Pfeil, B., Pierrot, D., Poulter, B., Rehder, G., Resplandy, L., Robertson, E., Rocher, M., Rödenbeck, C., Schuster, U., Schwinger, J., Séférian, R., Skjelvan, I., Steinhoff, T., Sutton, A., Tans, P. P., Tian, H., Tilbrook, B., Tubiello, F. N., van der Laan-Luijkx, I. T., van der Werf, G. R., Viovy, N., Walker, A. P., Wiltshire, A. J., Wright, R., Zaehle, S., and Zheng, B.: Global Carbon Budget 2018, Earth System Science Data, 10, 2141-2194, 2018.

Andrew, R. M.: Global $CO_2$ emissions from cement production, Earth System Science Data, 10, 195-217, https://doi.org/10.5194/essd-10-195-2018, 2018.

Boden, T. A., Marland, G., and Andres, R. J.: Global, Regional, and National Fossil-Fuel $CO_2$ Emissions, available at: http://cdiac.ornl.gov/trends/emis/overview_2014.html (last access: July 2017), Oak Ridge National Laboratory, U.S. Department of Energy, Oak Ridge, Tenn., USA, 2017.

BP: BP Statistical Review of World Energy June 2018, available at: https://www.bp.com/content/dam/bp/en/corporate/pdf/energy-economics/statistical-review/bp-stats-review-2018-full-report.pdf, last access: June 2018.

UNFCCC, 2018. National Inventory Submissions 2018. United Nations Framework Convention on Climate Change. Available at: http://unfccc.int/process/transparency-and-reporting/reporting-and-review-under-the-convention/greenhouse-gas-inventories-annex-i-parties/national-inventory-submissions-2018; accessed June 2018.

*It is somewhat unclear how the statistics shown in Table 3 and Fig. S4 have been calculated for the GreenLITE vs. WRF-Chem measurements in section 4.1. Have the data from all chords related to e.g. T1 been combined and then the statistics is derived, or has each chord been treated independent and the resulting statistics shown in Table 3 and Fig. S4 reflect the average across all chords?*

We agree with the reviewer that this point needs clarification. (PS: following the recommendation from Referee #3, we have split the previous Table 3 into Table 3 and Table S2. The previous Figure S4 is now Figure S6 in the revised manuscript)

We have added the following sentence to make it clearer:

"The statistics shown in Table 3, Table S2 and Figure S6 also indicate the ability of the models to reproduce the $CO_2$ at two urban in-situ stations (JUS & CDS) and the GreenLITE™ measurements. As for the GreenLITE™ data, we first compute the hourly averages of the observed and modeled $CO_2$ concentrations over all 15 chords for each transceiver (T1 and T2), and then calculate the respective statistics."

*The discussion of the results in section 4.2.2 regarding the spatial gradients between different chords of the GreenLITE observations and the simulated counterparts, as well as the corresponding discrepancy between observations and model results should at least mention the potential impact of turbulent eddies and thermals. Those are likely to form in a convectively unstable atmosphere, i.e. during summer, and are unlikely to be represented properly in the MYJ PBL scheme (a local closure model) deployed in the WRF-*

*Chem simulations (c.f. Xiao-Ming et al., 2010). Ref.: Hu, Xiao-Ming, John W Nielsen-Gammon, and Fuqing Zhang. 2010. "Evaluation of Three Planetary Boundary Layer Schemes in the WRF Model." Journal of Applied Meteorology and Climatology 49 (9): 1831–44. doi:10.1175/2010JAMC2432.1*

We greatly appreciate the reviewer's suggestion and fully agree that the vertical mixing associated with turbulent eddies and thermals plays an important role in the $CO_2$ transport and dispersion. The impact of insufficient vertical mixing, local eddy diffusion and entrainment flux under convective conditions reproduced by the local closure MYJ PBL scheme is a plausible explanation for the model-observation misfits. The revised manuscript has included the following discussion as suggested by the reviewer:

"Another potential source of measurement-model discrepancy is the atmospheric transport modeling as proposed in H2. According to previous studies (e.g. Hu et al., 2010), the turbulent eddies and thermals are unlikely to be reproduced properly by the local closure MYJ PBL scheme, which results in insufficient vertical mixing under convective (unstable) conditions, e.g. during summer. It may also indicate that the WRF-Chem model at a 1-km horizontal resolution cannot reproduce the fine-scale (sub-kilometer) $CO_2$ concentration features over a complex urban environment in Paris, as the analysis of JUS and CDS in-situ measurements has shown in Section 4.2.1."

Reference:

Hu, X. M., Nielsen-Gammon, J. W., and Zhang, F.: Evaluation of three planetary boundary layer schemes in the WRF model. Journal of Applied Meteorology and Climatology, 49(9), 1831-1844, 2010.

***Specific comments:***

*P1 L36: I suggest replacing "have been used" with e.g. "have been or will be used" as you are refering also to future satellites.*

Text changed as suggested.

*P7 L21: please rephrase "low atmosphere", e.g. "lower part of the atmosphere"*

Text changed as suggested. The modified text is as follows:

"It is well known that the lower part of the atmosphere is, on average, more stable in winter than in summer."

*P7 L38: "in some respect superior" this should be formulated clearer. What I see from Table 3 is that RMSE with the BEP model is always better for T1 than for JUS, and better for T1 than for CDS with one exception.*

To further clarify this point, the sentence has been refined as follows:

"The RMSE with the BEP scheme is within the range of 4.5 to 9.6 ppm for T1 which is substantially superior to those of JUS and CDS, with only one exception at CDS during summer when the value is slightly better for CDS than for T1."

*P8 L21, P8 L37, and P10 L19: please rephrase "The std values", e.g. "Standard deviations"*

Text changed as suggested.

*P9 L38: What is the difference between the first two of the three hypotheses? Is H2 meant to refer to only transport model deficiencies, excluding inaccuracies in emissions? This should be made clearer. Also it should be made clear at the end of section 4.2.2 which hypothesis remains the most probable one.*

Yes, Hypothesis 1 is about potential inaccuracies or uncertainties of the emission inventory for the Paris urban area, whereas Hypothesis 2 refers to the imperfect modeling of the atmospheric transport and dispersion of $CO_2$ over the complex urban area. We have modified the statement to make it clearer:

"• H2 The models fail in the description of $CO_2$ concentrations within the Paris city because of imperfect representations of atmospheric transport processes, excluding inaccuracies in emissions;"

Our analyses indicate that the model-GreenLITE™ discrepancy during the summer is more likely the consequence of the measurement noise and bias in some of the chords, whereas it is hard to fully rule out the possibility of impacts of the emission spatial structure and the atmospheric transport that have been discussed in section 4.2.2. Therefore, we tend to be more cautious to make such an assessment based on our current knowledge. We have added the following sentence in the conclusion and discussion section to address this point:

"Although it is not yet fully understood, several evidences suggest an increase of measurement noise and bias in some of the GreenLITE™ chords during the summer season, that must be resolved or reduced before assimilating the whole dataset into the $CO_2$ atmospheric inversion system that aims at retrieving urban fluxes."

---

## Author Comment (AC3) · 2 Oct 2019

We would like to thank Referee #3 for his/her thoughtful comments and detailed suggestions to our manuscript. In the following, we answer to the reviewer's comments and indicate the changes in the manuscript that were implemented as a consequence of the recommendations. The comments are in black and italic. Our answers are in blue and plain text.

**Anonymous Referee #3**

Lian et al., present long open-path spectroscopy measurements for the City of Paris from December 2015 to November 2016 in conjunction with in-situ observations from towers in and around Paris as well as WRF-Chem simulated observations from two different urban canopy schemes. It is assumed that the authors are using the GreenLITE measurements along with the in-situ tower observations to discern which WRF-Chem urban canopy scheme can best represent vertical mixing and transport in urban areas.

My main concern with this article is that the specific objective/conclusions of the paper are unclear. The authors have conducted a lot of work analyzing data from many components, but it is uncertain as to whether they have drawn any solid conclusions.

The objective, which I assume is using GreenLITE and the in-situ observations to evaluate WRF-Chem urban canopy configurations, should be more clearly stated in the Introduction and the title of the paper should be changed.

Without a clear narrative, the paper mainly comes across as a presentation of data which is difficult to evaluate as a reader. I do not recommend that this work be accepted for publication in ACP without substantial revisions to clarify scientific objectives/conclusions.

We thank the reviewer for his/her work and comments on our paper. To answer this criticism, we have added a few sentences in the introduction and conclusion sections that make clearer the objectives of the paper and the conclusions that were derived.

Urban areas are significant sources of fossil fuel  $CO_2$  emissions.  $CO_2$  measurements in urban areas are used in conjunction with atmospheric transport models and statistical inversion techniques to estimate city  $CO_2$ emissions. The novel GreenLITETM laser imaging system deployed in Paris provides a much wider spatial coverage of atmospheric  $CO_2$  concentrations over the complex urban environment, which makes it possible to provide new insights into the  $CO_2$  characteristics compared to the highly accurate in-situ measurements that can only be made at point locations, and can be influenced by local sources to a poorly known extent. In this paper, we analyze the measurements provided by this novel system, together with the more classical in-situ sampling and high-resolution modeling and we focus on the temporal and spatial variability of atmospheric  $CO_2$  concentrations. The main purpose of the paper is therefore an evaluation of the new system capabilities to provide information on both the emission and the atmospheric transport, typically whether the new system can falsify the emission inventory used or point out to transport modeling deficiencies (the two hypotheses formulated in the paper).

We have added the following paragraph in the introduction to better clarify the objectives of this work:

"The detailed objectives of this paper are:

- To analyze in detail the information content of the GreenLITE™ data in addition to conventional in-situ CO2 measurements in order to better understand the temporal and spatial variations of near-surface CO2 concentrations over Paris and its vicinity.

- To evaluate the performance of the high-resolution WRF-Chem model coupled with two urban canopy schemes (UCM, BEP) for the transport of CO2 over the Paris megacity area based on the two types of CO2 measurements.
- To discuss the potential implications of assimilating the GreenLITE™ data into the CO2 atmospheric inversion systems with the ultimate goal of increasing the robustness of the quantification of city emissions and constraining the spatial distribution of the emissions within the urban area."

The main conclusions of this study are:

- Two urban canopy schemes (UCM, BEP) as part of the WRF-Chem model are capable of reproducing the seasonal cycle and most of the synoptic variations in the atmospheric CO2 in-situ measurements over the suburban areas, as well as the general corresponding spatial differences in CO2 concentration between pairs of in-situ stations that span the urban area.
- The GreenLITETM measurements are less sensitive to local unresolved sources than the in-situ point measurements, and are then better suited for the comparison to km-scale modeling. In our analysis, the GreenLITETM data have been used to show a deficiency of the UCM scheme during the winter, linked to underestimated vertical mixing. Conversely, the model-GreenLITETM discrepancy that is observed during the summer is not yet fully understood. Several evidences suggest an increase of measurement noise and bias in some of the GreenLITETM chords during the summer season, that must be resolved or reduced before assimilating the whole dataset into the CO2 atmospheric inversion system that aims at retrieving urban fluxes.
- Within the city, the misfit between the observed and simulated CO2 concentrations is found to be highly sensitive to the WRF-Chem configuration for the urban canopy scheme, which affects the atmospheric vertical mixing. We also show that the CO2 concentrations are impacted by the spatial distribution of the emission and the presence of local sources that are poorly resolved in the inventory. This study stresses the difficulty in reproducing precisely the atmospheric CO2 concentration within the city because of our inability to represent the detailed spatial structure of the emission and because of the sensitivity of the concentration to the strength of vertical mixing. From the model results analysis, we infer that the uncertainty on the vertical mixing is much larger than the uncertainty on the emissions so that atmospheric concentration measurements within the city can hardly be used to constrain the emission inventories.

We have modified the conclusion and discussion section based on the main conclusions listed above.

I have the following suggestions regarding the technical components of the analysis.

**Main Comments:**

(1) The observational network: can you provide some indication as to areas in which the observations are most sensitive especially CDS, JUS, and any others that are situated in or adjacent to major sources/sinks? I understand that the authors use WRFCHEM and not a Lagrangian approach so that footprints cannot be generated but having some understanding would help the comparison of GreenLITE and in-situ observations presented later in the paper since these two different types of measurements represent different spatial extents.

The footprint of the measurement station very much depends on the wind speed and direction, as well as the atmospheric stability. It is then difficult to interpret a mean footprint that aggregates a wide range of atmospheric situations. In response to the reviewer's concern about the major sources of emissions to the station, we carried out some sensitivity experiments for the one-month period of March 2016 with

anthropogenic and biogenic emissions limited to a given area within the simulation domain in order to quantify their respective contributions to the simulated CO2 concentrations at a certain measurement site. This set of experiments includes the assignment of emissions to: 1) ONE: one grid cell that contains the insitu station, 2) GRP: all grid cells within the GReater Paris except the one where the station is located, 3) IDF: all grid cells within the IdF region except those of the Greater Paris, 4) OUT: all grid cells outside the IdF region, as shown in Figure S10 (a) (b) (c) (d) respectively. The contribution from sources outside the model domain is small enough so that its influence is negligible. Figure S11 shows the relative contributions (in percentages) of each component to the modeled total anthropogenic and biogenic CO2 concentrations for one urban site JUS and one suburban site COU respectively. The simulated monthly mean concentrations of anthropogenic CO2 are 11.0 ppm at JUS and 5.4 ppm at COU, which are much larger than those of biogenic CO2 (0.6 ppm at JUS and 0.7 ppm at COU). In general, an urban station like JUS is under a strong influence of the anthropogenic emissions within the IdF region. The contributions of anthropogenic emissions in the vicinity of the station (ONE) and from the Greater Paris (GRP) areas to the simulated anthropogenic CO2 concentrations are around 16% and 60% respectively, whereas the remote anthropogenic emissions account for less than 20%. For a suburban station like COU, the Parisian emissions (GRP) and the remote ones (OUT) have a comparable influence (~40%) on the simulated anthropogenic concentrations, with very large variations depending on the wind direction (downwind or upwind of the city). The biogenic  $CO_2$  concentrations mainly come from outside of the IdF region (~86%).

We have added the following statement in the main body of the revised manuscript and two figures (Figure S10 and S11) in the supplement based on the discussion above:

"Atmospheric transport simulations make it possible to assess the respective contributions of various areas/sectors to the measurements. Our preliminary sensitivity experiments (see Figure S10 and S11 for details) have shown that the anthropogenic emission from the Greater Paris area is the dominant contribution (~80%) to the anthropogenic CO2 signal at the urban measurement stations. In order to get further insights into the characteristics of CO2 spatial variations within the Paris city, it is therefore necessary to analyze the CO2 differences with the consideration of the anthropogenic CO2 emissions shown in Figure 2 and Figure 3."

For better clarity, we have also added the following analyses in the supplement together with Figure S10 and S11.

"In order to determine respective contributions of various areas/sectors to the simulated  $CO_2$  concentrations at a certain measurement site, we carried out a set of sensitivity experiments for the one-month period of March 2016 with anthropogenic and biogenic emissions limited to a given area. This set of experiments includes the assignment of emissions to: 1) ONE: one grid cell that contains an in-situ station, 2) GRP: all grid cells within the GReater Paris except the one where the station is located, 3) IDF: all grid cells within the IdF region except those of the Greater Paris, 4) OUT: all grid cells outside the IdF region, as shown in Figure S10 (a) (b) (c) (d) respectively.

Figure S11 shows the relative contributions (in percentages) of each component to the modeled total anthropogenic and biogenic  $CO_2$  concentrations for one urban site JUS and one suburban site COU respectively. The simulated monthly mean concentrations of anthropogenic  $CO_2$  are 11.0 ppm at JUS and 5.4 ppm at COU, which are much larger than those of biogenic  $CO_2$  (0.6 ppm at JUS and 0.7 ppm at COU). In general, an urban station like JUS is under a strong influence of the anthropogenic emissions within the IdF region. The contributions of anthropogenic emissions in the vicinity of the station (ONE) and from the Greater Paris (GRP) areas to the simulated anthropogenic  $CO_2$  concentrations are around 16% and 60% respectively, whereas the remote anthropogenic emissions account for less than 20%. For a suburban station

like COU, the Parisian emissions (GRP) and the remote ones (OUT) have a comparable influence (~40%) on the simulated anthropogenic concentrations, with very large variations depending on the wind direction (downwind or upwind of the city). Note that in these experiments, the emission inventory and the WRF-Chem modeling cannot describe the CO2 patterns (both emission and concentration) at a scale finer than 1 km, and the simulation shows that the "local" contribution is significant. The unresolved spatial distribution of the emission can therefore be a significant contribution to the uncertainty. The biogenic CO2 concentrations mainly come from outside of the IdF region (~86%)."

Figure S10. Four experiments are carried out for the JUS station with the assignment of emissions to: (a) ONE: one grid cell that contains an in-situ station; (b) GRP: all grid cells within the Greater Paris except the one where the station is located; (c) IDF: all grid cells within the IdF region except those of the Greater Paris; (d) OUT: all grid cells outside the IdF region. Another four experiments are carried out for the COU station.

Figure S11. Relative contributions (in percentages) of each component flux to the modeled total anthropogenic and biogenic CO2 concentrations for (a) urban site JUS and (b) suburban site COU. Note that only the afternoon data (11-16 UTC) are used in the analysis.

**(2) GreenLITE campaign – can you provide more description than citing the Zaccheo paper as to how the GreenLITE observations are calibrated?**

As mentioned by the reviewer, the calibration method is extensively described in Zaccheo et al. (2019). Nevertheless, we have followed the suggestion and now provide a better description (although very much summarized) of the procedure:

"These slowly time-varying differences were most likely due to a slight systematic long-term drift in both the on- and off-line wavelengths as a function of continuous operations. Such drift may induce some nonlinear impacts on the measured concentrations. It is therefore more appropriate to adjust the wavelengths rather than to apply a linear calibration to the retrieved concentrations. Unlike in-situ point measurement systems, there is no established method for calibration of long open-path systems to the WMO mole fraction scale used as an international standard for atmospheric  $CO_2$  monitoring (Tans et al., 2011). Therefore, a bias correction method was developed by AER (Zaccheo et al., 2019) for addressing observed slowly drifting biases between the GreenLITETM prototype system and the two in-situ sensors (CDS and JUS) that are near the GreenLITETM chords. This method computed a time-varying adjustment to the offline wavelength based on a non-linear optimization mechanism. This non-linear approach adjusts the GreenLITETM offline wavelength considering not only the average values of hourly CO2 concentrations at two in-situ stations, but also the corresponding average temperature, relative humidity, atmospheric pressure along the chord and an optimized online wavelength value during the measurement period. Finally, the median on- and off-line values over a 4-day window was used to recompute the GreenLITE™ data from all chords using a radiative transfer based iterative retrieval scheme based on the LBLRTM model (Clough et al., 2005). Even though this approach is not ideal as the two in-situ stations and the GreenLITETM system do not sample the exact same area, it does provide a well-defined mechanism that reduces the systematic long-term biases with no significant impact on the chord-to-chord variations."

A general response to comments (3) (4) and (5):

We fully agree with the reviewer that the uncertainties associated with atmospheric transport, anthropogenic emissions, biogenic fluxes and background conditions could all have a more or less impact on the model performance. Over the years, many studies have been carried out at different scales and regions on the analysis, derivation and quantification of critical sources of uncertainties that lead to the model-observation misfits. Nevertheless, even with a state-of-art atmospheric transport model and an inventory at a high spatio-temporal resolution, the uncertainties associated with the modeling are inevitable and cannot be completely eliminated. The ensemble-based sensitivity study and a full analysis on the uncertainties derived from different anthropogenic inventories, biogenic fluxes, atmospheric transports or background conditions are out of the scope of this study that focuses on the potential contribution of GreenLITETM observing system, but will be specifically addressed within another dedicated study.

(3) WRF-Chem – Is this paper an analysis of WRF-Chem urban canopy models for cities like Paris (evaluated using GreenLITE and in-situ observations)? If so, please substantiate/provide reference for the claim that "previous sensitivity tests indicate that different physical schemes in the WRF-Chem model lead to mean differences of 2-3ppm on the simulated  $CO_2$  concentrations over Paris, whereas the various urban canopy schemes lead to much larger differences." This seems like the motivation for much of the work presented within the paper, but I am not sure that this claim, if substantiated, holds true across most urban areas.

As our answers to the general comment, our main objective is not fully to test the modeled CO2 sensitivity to the use of different physical schemes and to discern which urban canopy scheme could reach best results when comparing the model to observations. Several studies have demonstrated that the city-scale physical and dynamical processes in the atmospheric modeling system remain a challenge. In order to select an adequate WRF-Chem model configuration for Paris, we did perform some preliminary sensitivity experiments to test the impact of different physical schemes (5 PBL schemes + UCM, 2 PBL schemes + BEP) on the simulated CO2 concentrations. The simulations were carried out for two months, including one winter month (January 2016) and one summer month (July 2016). These preliminary sensitivity results indicate that different PBL schemes in the WRF-Chem model lead to monthly average differences of 2-3ppm on the simulated CO2 concentrations over Paris, whereas the urban canopy schemes lead to much larger differences of 8-10 ppm. We thus carried out the 1-year simulation with two different urban canopy schemes as they are sufficient to address the paper main question regarding the ability of a configuration of the WRF-Chem model to simulate the CO2 atmospheric transport in an urban environment, but also to provide an estimate of the modeling uncertainty.

We have added the following sentence in the manuscript to account for the reviewer's comment:

"In order to select an adequate model physical configuration for Paris, we carried out some preliminary sensitivity experiments to test the impact of different physical schemes on the simulated  $CO_2$  concentrations. These tests use up to five different PBL schemes and two urban canopy schemes. The simulations were carried out for two months, including one winter month (January 2016) and one summer month (July 2016). These preliminary sensitivity results indicate that different PBL schemes in the WRF-Chem model lead to monthly average differences of 2-3 ppm on the simulated  $CO_2$  concentrations over Paris, whereas the two different urban canopy schemes lead to much larger differences of 8-10 ppm. Thus in this study, we carried out the 1-year simulation with two different urban canopy schemes as they are sufficient to address the paper main question regarding the ability of a configuration of the WRF-Chem model to simulate the  $CO_2$  atmospheric transport in an urban environment, but also to provide an estimate of the modeling uncertainty. All of the other physics options remained the same for the two experiments (Table 2)."

a. If this is not the focus, and the purpose is to use the meteorology to understand the variability of the measurements, then I believe the authors should pick a model and use it throughout the rest of the analysis. It seems (from Figure 5) that the BEP model is largely better. The rest of the analysis using UCM could be put in the supplementary information. As an aside, I do think that the authors could use the ensembles in a way that would help them draw some robust conclusions. Their ensembles provide some measure of the atmospheric transport and dispersion uncertainty which can be used to contextualize their comparison between GreenLITE and the in-situ observations (S1 and S4).

The main purpose of the study is to assess the potential contribution of the GreenLITETM system, in addition to the more classical in-situ sampling, for a better understanding of the temporal and spatial variations of near-surface CO2 concentrations over Paris and its vicinity (due to emissions / atmospheric transport). Even though the two urban canopy schemes do not represent the full range of uncertainty in the atmospheric transport, in some extent they do provide an insight into the critical impact of the atmospheric transport on simulated atmospheric CO2 concentrations. It thus appears to us that it is necessary to keep the UCM analysis within the main body of the paper. We also tested different PBL schemes for the impact of the atmospheric transport, but decided to show results for the two urban canopy schemes because they caused the largest differences in simulated CO2 concentrations.

(4) Anthropogenic Fluxes – The use of the anthropogenic fluxes in the analysis should be reconsidered or better explained. For example, does IER have any temporal variability? If so, please explain. If not, the authors could consider scaling using published methods. The authors could use other emission products that have temporal variability if needed. The loss of spatial scale (e.g. going from 5-10km) seems less important than preserving some temporal structure in emissions. Other products are also more recent and thus more represented of ex-urban fluxes which constitute a large portion of  $CO_2$  inflow to Paris.

Yes, the IER inventory used in this study has a detailed country-specific temporal profiles (monthly, daily and hourly) at spatiotemporal resolutions of 5 km and 1 h. Given the fact that it has a higher spatial resolution than some other emission products and it has been rescaled to account for annual changes in emission between the base year and simulation timeframe, this inventory is sufficient to be used in this study.

We have added the information about the IER temporal variability in the revised manuscript:

"CO2 emissions from fossil fuel CO2 sources outside the IdF region are taken from the inventory of the European greenhouse gas emissions, together with country-specific temporal profiles (monthly, daily and hourly) at a spatial resolution of 5 km (updated in October 2005). This inventory was developed by the Institute of Economics and the Rational Use of Energy (IER), University of Stuttgart, under the CarboEurope-IP project (http://www.carboeurope.org/)."

Could the authors also further explain "we interpolate the emissions to the WRF-Chem grids following the principle of mass conservation?" This is unclear in both its meaning and why it is important.

The total magnitude of anthropogenic emissions should be consistent before and after the interpolation to model grid cells. We have modified the statement to make it clearer:

"Finally, we interpolate the emissions onto the WRF-Chem grids, making sure to conserve the total budget of emission in the process, as done in previous studies (e.g. Ahmadov et al., 2007)."

As with the WRF-Chem comments, the authors could use an ensemble of anthropogenic emission products (those outside of Paris) to help contextualize the GreenLITE and in-situ observations in terms of emission uncertainty (refer to Martin et al., 2018).

Please see the analysis above. The contribution of anthropogenic emissions outside the IdF region to the simulated anthropogenic  $CO_2$  concentrations over urban areas is relatively small (~20%), and our present simulations are capable of reproducing the seasonal cycle and most of the synoptic variations in the atmospheric  $CO_2$  point measurements over the suburban areas. In addition, this distant contribution is much smoother, both in temporal and spatial scales, than the impact of more local emissions. We then do not expect a significant impact of these distant emissions on the  $CO_2$  signatures that are analyzed in the paper. We then do not feel that an ensemble of anthropogenic emission products outside the IdF region will bring critical insights on the main conclusions at two urban in-situ stations (JUS & CDS) and the GreenLITETM measurements. A deeper analysis of the impact of uncertainties in the anthropogenic emissions outside the IdF region so utside the IdF region so utside the IdF region so utside the IdF region so the main conclusions at two urban in-situ stations (JUS & CDS) and the GreenLITETM measurements. A deeper analysis of the impact of uncertainties in the anthropogenic emissions outside the IdF region so utside the IdF region is out of the scope of the paper.

(5) Biogenic Fluxes – The use of VPRM to represent the urban biosphere is an active area of research and there are lots of questions as to how well a biospheric model captures the urban biogenic emissions. When VPRM was optimized using flux data, were urban towers used to help parameterize the "urban" areas of Paris? The paper mentions that the western portion of Paris has much green space and thus biogenic sources might be important in this area of the city and impact the analysis. How was Paris-VPRM (or VPRM) validated, e.g. comparison to in-situ data from towers outside of the city that are surrounded by vegetation (maybe OVS)? Has it been used in other studies? How does it vary as a function of time in comparison to the anthropogenic fluxes like what is shown in Figure 3?

As for the urban biogenic emissions mentioned by the reviewer, we certainly agree that it might be important for the simulated  $CO_2$  concentrations and it is an active area of research. To which extent the biogenic fluxes affect the simulated  $CO_2$  concentrations in the Paris urban areas remains an open question. Whereas there is no eddy covariance measurement in the Paris urban area that is available for the biospheric flux optimization and we are not able to make an evaluation of the Paris-VPRM model in this study. Nevertheless, we have performed some further analyses and validations of the VPRM model at a suburban station at SAC in a dedicated study mentioned above. Since these analyses at SAC do not reflect the model performance of the biosphere mode in the urban area, it is out of the scope of this study. Mean diurnal cycles of  $CO_2$  biogenic flux (NEE) for 12 calendar months and for 8 vegetation classes used in VPRM over Domain 03 are shown in Figure 4 and the related texts are in Section 3.2.2.

(6) Results – (4.1) There are a lot of moving pieces in this analysis and it is hard to ascertain the main conclusions from the statistical analysis. Do you think that the uncertainties associated with the other components (e.g. anthropogenic emissions and vprm sources and sinks) would have changed some of these results especially during the growing seasons or per your analysis of the seasonality of the sectors? From the Table, it is unclear that BEP outperforms UCM for much of the year. As with 4.1, I am not sure what the main takeaway is from this analysis.

In fact, each paragraph in Section 4.1 relates to a certain aspect regarding the statistics for observed and modeled CO2 concentrations for periods of the day (all hourly data, hourly afternoon data) and two urban canopy schemes (UCM, BEP). In general, the model performance is better during the afternoon than it is for the full day. UCM and BEP have different performances for four seasons and for urban/suburban areas (see answers below, this is also the main takeaway). The statistics further confirm the fact that the GreenLITETM measurements represent an average over a wide area, and are then less sensitive to local unresolved sources than the in-situ measurements.

We do not know to what extent the uncertainties associated with the other components (e.g. anthropogenic emissions and VPRM sources and sinks) would have changed some of these results since we have not made the relevant sensitivity experiments. A full analysis of these uncertainties would be a paper by itself.

In the third paragraph of Section 4.1, we have already discussed the different performances of UCM and BEP for four seasons and for urban/suburban areas with the following statements: 1) The statistics for BEP compared to the observations within the urban areas are significantly better than UCM during autumn and winter; 2)  $CO_2$  concentrations are better reproduced by both UCM and BEP in the spring; 3) Both models show lower correlations during summer; 4) the UCM and BEP also have comparable performances at periurban areas while the BEP is slightly better at some suburban sites as shown by the statistics.

(4.2.1) Why did you use the wind per ECMWF versus wind measurements at the upwind tower(s)? I am sure, on average, the ECMWF winds are similar to what is measured at the towers but since you are comparing hourly measurements, this may make a difference.

The ECMWF wind product is used here for 2 reasons: Firstly, our previous study has shown that the wind speeds provided by the ECMWF high-resolution operational forecasts (HRES) are, in general, closer to the observations than those provided by WRF (Lian et al., 2018). Secondly, the WRF model was run with two configurations (UCM and BEP urban canopy schemes) in this study. If we make use of the modeled winds, the UCM and BEP modeled  $CO_2$  spatial differences should be analyzed using their corresponding modeled wind fields, and the observed winds are then needed for the analysis of the observed  $CO_2$  spatial differences. However, given the small-scale wind variations reproduced by the model, it is hard to determine that the wind data at which station should be used in the analysis. For the purpose of a fair and uniform comparison, we thus use an independent wind product. The HRES with a horizontal resolution of about 16 km could provide a synoptic wind pattern as a proxy for all stations located within the IdF region. We have added the following sentence in the manuscript to account for the reviewer's comment:

"The HRES wind product is used here for two reasons: Firstly, our previous study has shown that the wind speeds provided by HRES are, in general, closer to the observations than those provided by WRF (Lian et al., 2018). Secondly, the WRF-Chem model was run with two configurations (UCM and BEP urban canopy schemes) in this study. If we make use of the modeled winds, the UCM and BEP modeled  $CO_2$  spatial differences should be analyzed using their corresponding modeled wind fields, and the observed winds are then needed for the analysis of the observed  $CO_2$  spatial differences. However, given the small-scale wind variations reproduced by the model, it is hard to determine that the wind data at which station should be used in the analysis. For the purpose of a fair and uniform comparison, we thus use an independent wind product."

**Also, how much time does it take to traverse some of the towers that are farther apart (e.g. COU and SAC)? Did you compare observations from similar times or did you account for a lag in the measurements via travel time?**

We ignore the time lag needed to transport information from upwind to downwind sites spanning the city by computing spatial gradients between concentrations at a given time. This is mainly due to the fact that the consideration such a time lag might be somewhat meaningless given the wind shear in the PBL during the afternoon when the mixing layer is usually well developed. Typical wind speed over Paris at 700 m above ground level is 7 m/s (25 km/h) and the distance between COU and SAC is approximately 38 km so that air masses take, on average, less than 2 hours to travel between the two sites at this height. Conversely, the wind speed at ground level is much smaller so that there is not a single time-lag that can be used. We thus assume that the analysis that is based on  $CO_2$  concentration differences measured during the same 1hour window is a minor issue. Note that, when comparing observation and models, the time lags are consistent.

**Minor Comments:**

Be specific as to what model you are using. I think in most cases you are referring to WRF-Chem models but there are others too such as VPRM, etc.

Following the reviewer's recommendation, we have attempted to make it clear and specific.

Grammar should be checked in many places throughout the article to improve clarity. Examples include lines 34 through 36 (page 2),  $\sim 10$  (page 8).

As suggested, we have carefully done thorough English editing and corrected the grammatical mistakes in the revised manuscript.

Figures should be modified to improve clarity:

For example, Figure 1 should include a depiction of adjacent urban areas to show how remote AND, COU, OVS, and SAC are from ex-urban sources. This will help the reader know whether or not they sample "clean" air.

We feel that there is no need to add a depiction in Figure 1 for 3 reasons: 1) the distributions of the 1-km anthropogenic  $CO_2$  emissions together with all in-situ measurement stations are shown in Figure 2a. 2) the dominant land use categories together with all in-situ measurement stations are shown in Figure S5. These two figures could be sufficient to provide such a depiction. 3) As we have mentioned in Section 2.1, the insitu stations are installed on the rooftops or on towers to minimize the impact of local surface emissions. Moreover, the distance to the localized emissions was also taken into account as a necessary aspect in the design of this  $CO_2$  monitoring network to ensure they sample air that is not in the immediate proximity of large anthropogenic emissions.

Figure 2 should include roads and other infrastructure in the second panel especially since the authors have made spent time discussing sectorial emissions. Also note (a) and (b) on Figure 2.

This suggestion is well taken. We have added these infrastructures into a second panel in Figure 1 where it might be more appropriate.

As suggested, we have noted (a) and (b) on Figure 2.

For Figure 4, zoom into similar area as in Figure 2 to show if VPRM is capturing urban biospheric flux which can significantly impact the urban fluxes especially their variability.

Figure R1 a high-resolution zoom of Paris and shows the daytime (06-18 UTC) average of  $CO_2$  biogenic flux (NEE) in June 2016. Due to the 1-km SYNMAP land use data used for the VPRM model, the biogenic fluxes in Paris are almost zero except for a few grid cells containing two big parks that are located in the eastern and western Paris.

We thus feel that there is no need to make this high-resolution zoom of the GreenLITETM covering areas since we have already mentioned it in Section 3.2.2 "The model simulates negative values of NEE (uptake of more than 5 gCO2/m2/day) over most of the region with the exception in urban areas where the values are assigned to zero."